# SCF (Fbxl17) ubiquitylation of Sufu regulates Hedgehog signaling and medulloblastoma development

Madalina Raducu[1], Ella Fung[1], Sébastien Serres[1], Paola Infante[2], Alessandro Barberis[1], Roman Fischer[3], Claire Bristow[1], Marie-Laëtitia Thézénas[3], Csaba Finta[4], John C Christianson[5], Francesca M Buffa[1], Benedikt M Kessler[3], Nicola R Sibson[1], Lucia Di Marcotullio[6,7], Rune Toftgård[4] & Vincenzo D'Angiolella[1,*]

## Abstract

Skp1-Cul1-F-box protein (SCF) ubiquitin ligases direct cell survival decisions by controlling protein ubiquitylation and degradation. Sufu (Suppressor of fused) is a central regulator of Hh (Hedgehog) signaling and acts as a tumor suppressor by maintaining the Gli (Glioma-associated oncogene homolog) transcription factors inactive. Although Sufu has a pivotal role in Hh signaling, the players involved in controlling Sufu levels and their role in tumor growth are unknown. Here, we show that Fbxl17 (F-box and leucine-rich repeat protein 17) targets Sufu for proteolysis in the nucleus. The ubiquitylation of Sufu, mediated by Fbxl17, allows the release of Gli1 from Sufu for proper Hh signal transduction. Depletion of Fbxl17 leads to defective Hh signaling associated with an impaired cancer cell proliferation and medulloblastoma tumor growth. Furthermore, we identify a mutation in Sufu, occurring in medulloblastoma of patients with Gorlin syndrome, which increases Sufu turnover through Fbxl17-mediated polyubiquitylation and leads to a sustained Hh signaling activation. In summary, our findings reveal Fbxl17 as a novel regulator of Hh pathway and highlight the perturbation of the Fbxl17–Sufu axis in the pathogenesis of medulloblastoma.

**Keywords** F-box protein; Fbxl17; Hedgehog signaling; medulloblastoma; Sufu
**Subject Categories** Cancer; Post-translational Modifications, Proteolysis & Proteomics; Signal Transduction
The EMBO Journal (2016) 35: 1400–1416

## Introduction

Signals of the Hh (Hedgehog) pathway are essential in directing cell proliferation and patterning during embryogenesis (Hooper & Scott, 2005; Ingham *et al*, 2011). The pathway lays at the cross-road of cell survival and differentiation decisions, and unscheduled Hh signaling can lead to hyperproliferation and the development of malignancies. Indeed, alterations of pathway components have been described in multiple cancer types (Teglund & Toftgard, 2010).

Hh signaling is initiated at the cell membrane by the inactivation of the 12-pass transmembrane receptor Ptch (Patched) upon binding of the Hh ligands (Ingham *et al*, 2011). Ptch inactivation frees the cognate receptor Smo (Smoothened) leading to the downstream activation of a transcriptional program mediated by Gli1, Gli2, and Gli3 (Glioma-associated oncogene homolog 1, 2, and 3) transcription factors (Hui & Angers, 2011).

Sufu (Suppressor of fused) acts as a central negative regulator of Hh signaling by sequestering the Gli transcription factors in an inactive complex (Kogerman *et al*, 1999; Dunaeva *et al*, 2003; Merchant *et al*, 2004; Humke *et al*, 2010). Sufu deletion in mice leads to the continuous activation of Hh signal and embryonic lethality at day 9.5 (Cooper *et al*, 2005; Svard *et al*, 2006), underscoring the essential role of Sufu for proper development. Importantly, heterozygous loss of Sufu, in conjunction with the loss of p53, leads to the development of medulloblastoma and rhabdomyosarcoma (Lee *et al*, 2007). Germline Sufu mutations have been identified in medulloblastoma (Taylor *et al*, 2002; Kool *et al*, 2014) and associated with the development of medulloblastoma in Gorlin syndrome (Pastorino *et al*, 2009; Kijima *et al*, 2012; Smith *et al*, 2014). Furthermore, somatic Sufu mutations have been identified in multiple other malignancies, including prostate cancer (Sheng *et al*, 2004).

1  Cancer Research UK and Medical Research Council Institute for Radiation Oncology, Department of Oncology, University of Oxford, Oxford, UK
2  Center for Life NanoScience@Sapienza, Istituto Italiano di Tecnologia, Rome, Italy
3  Target Discovery Institute, Nuffield Department of Medicine, University of Oxford, Oxford, UK
4  Department of Biosciences and Nutrition, Center for Innovative Medicine, Karolinska Institutet, Huddinge, Sweden
5  Ludwig Institute for Cancer Research, University of Oxford, Oxford, UK
6  Department of Molecular Medicine, University "La Sapienza", Rome, Italy
7  Pasteur Institute/Cenci Bollognetti Foundation, Sapienza University, Rome, Italy
   *Corresponding author. Tel: +44 01865617400; E-mail: vincenzo.dangiolella@oncology.ox.ac.uk

Sufu has a pivotal role in Hh signaling and is a relevant tumor suppressor, but the players involved in controlling Sufu levels are unknown and the relevance for cancer proliferation remains unrecognized.

Using an unbiased proteomic approach, we have identified Sufu as an interacting partner of the E3 ubiquitin ligase Fbxl17 (F-box and leucine-rich repeat protein 17). Fbxl17 belongs to a family of E3 ubiquitin ligases containing a leucine-rich repeat and an F-box motif, which mediate binding to Skp1 (S-phase kinase-associated protein 1) (Skaar *et al*, 2009). Skp1 in turn recruits Cul1 (Cullin 1) and Rbx1 (RING-box protein 1) for the formation of an SCF (Skp1-Cul1-F-box protein) complex. The multi-subunit SCF directs the polyubiquitylation of target substrates by recruiting an E2 enzyme with an activated ubiquitin molecule ready to be transferred on accepting lysines of target substrates. Substrate specificity is dictated by the presence of variable protein–protein interaction domains within diverse F-box proteins (Skaar *et al*, 2013). According to the type of protein–protein interaction domain, three subfamilies of F-box proteins have been identified in the human genome: Fbxws (containing WD40 repeats), Fbxls (containing leucine-rich repeats, LRR), and Fbxos (containing other interacting motifs) (Jin *et al*, 2004).

F-box proteins can function as tumor suppressor or oncogenes and have been implicated in the control of cell proliferation and cancer development (D'Angiolella *et al*, 2012; Bassermann *et al*, 2014). Novel compounds targeting F-box proteins have been developed, and therefore, F-box proteins represent candidate pharmacological targets to treat cancer (Skaar *et al*, 2014). However, few F-box proteins have been matched to substrates and biological roles.

Here, we show that Fbxl17 forms a functional SCF complex and targets the tumor suppressor Sufu for proteolysis in the nuclear compartment upon Hh pathway activation. The ubiquitylation of Sufu, mediated by Fbxl17, allows the release of Gli1 for full activation of the Hh signaling. Extending our basic biological finding to the significance of the Fbxl17–Sufu axis *in vivo*, we have identified Fbxl17 as a crucial regulator of cell proliferation in medulloblastoma.

Finally, we show that in human disease, a somatic mutation in Sufu, occurring in medulloblastoma of patients with Gorlin syndrome, increases Sufu turnover through Fbxl17 polyubiquitylation, leading to a sustained Hh signaling and cell proliferation. Our study highlights the alteration in Fbxl17–Sufu axis as an etiological mechanism of medulloblastoma.

## Results

### Fbxl17 interacts with Sufu and regulates Sufu levels

To identify the putative functions and substrates of Fbxl17, we isolated Fbxl17 and interacting proteins from HEK293T cells. Liquid chromatography–tandem mass spectrometry (LC-MS/MS) of the immunoprecipitated material identified peptides corresponding to components of the SCF complex (Skp1 and Cul1), peptides corresponding to BACH1 (BTB and CNC homology 1 protein, a known substrate of Fbxl17) (Tan *et al*, 2013) as well as putative novel substrates such as Sufu and ZBTB33 (Zinc Finger and BTB domain containing 33) (Table 1).

Since a single substrate might be targeted by different ligases (like in the case of β-TrCP1 and β-TrCP2; beta-transducin repeat containing E3 ubiquitin protein ligase 1 and 2) (Guardavaccaro *et al*, 2003), we screened a panel of Flag-tagged F-box proteins isolated from HEK293T cells for the binding to Sufu. We detected Sufu only in Fbxl17 co-immunoprecipitates (Fig 1A), whereas all F-box proteins bound to Skp1.

**Table 1.  Proteins associated with Fbxl17 identified by mass spectrometry.**

| Accession | Score | Mass | No. matches | No. sig. matches | No. seq | No. sig. seq | emPAI | Description |
|---|---|---|---|---|---|---|---|---|
| Q9UF56 | 33,624 | 77,984 | 1,316 | 1,081 | 76 | 74 | 178.17 | Fbxl17 |
| P63208 | 8,032 | 18,817 | 297 | 243 | 17 | 15 | 59.82 | Skp1 |
| Q13616 | 248 | 90,306 | 14 | 12 | 9 | 7 | 0.28 | Cul1 |
| Q9UMX1 | 112 | 54,255 | 9 | 5 | 3 | 3 | 0.27 | Sufu |
| Q96DT7 | 102 | 95,975 | 6 | 3 | 4 | 3 | 0.11 | ZBTB10 |
| Q9BY89 | 79 | 197,617 | 9 | 4 | 9 | 4 | 0.07 | KIAA1671 |
| Q86T24 | 65 | 75,065 | 1 | 1 | 1 | 1 | 0.04 | ZBTB33 |
| O14867 | 59 | 83,844 | 2 | 1 | 2 | 1 | 0.04 | BACH1 |
| O43318 | 55 | 67,895 | 4 | 2 | 4 | 2 | 0.10 | TAB 1 |
| O43164 | 39 | 79,192 | 3 | 1 | 3 | 1 | 0.04 | Praja2 |
| Q04721 | 21 | 279,082 | 5 | 2 | 3 | 1 | 0.01 | Notch 2 |
| Q9BXU1 | 20 | 116,647 | 7 | 1 | 3 | 1 | 0.03 | SgK396 |
| O14641 | 17 | 79,184 | 5 | 1 | 5 | 1 | 0.04 | DSH homolog 2 |
| O15355 | 16 | 59,919 | 9 | 2 | 6 | 2 | 0.11 | PPM1G |

The table represents the summary of four independent 3×Flag-tagged Fbxl17 immunoprecipitation experiments followed by liquid chromatography–mass spectrometry (LC/MS). Identified interacting proteins in Fbxl17 were filtered using LC/MS data from 8 independent analyses of unrelated F-box proteins 3×Flag immunoprecipitations. Results were scored according to confidence of identification and a Mascot Mowse cutoff score of 50 was established. Proteins identified with a Mascot cutoff score below 50 in more than one biological replicate are also reported.

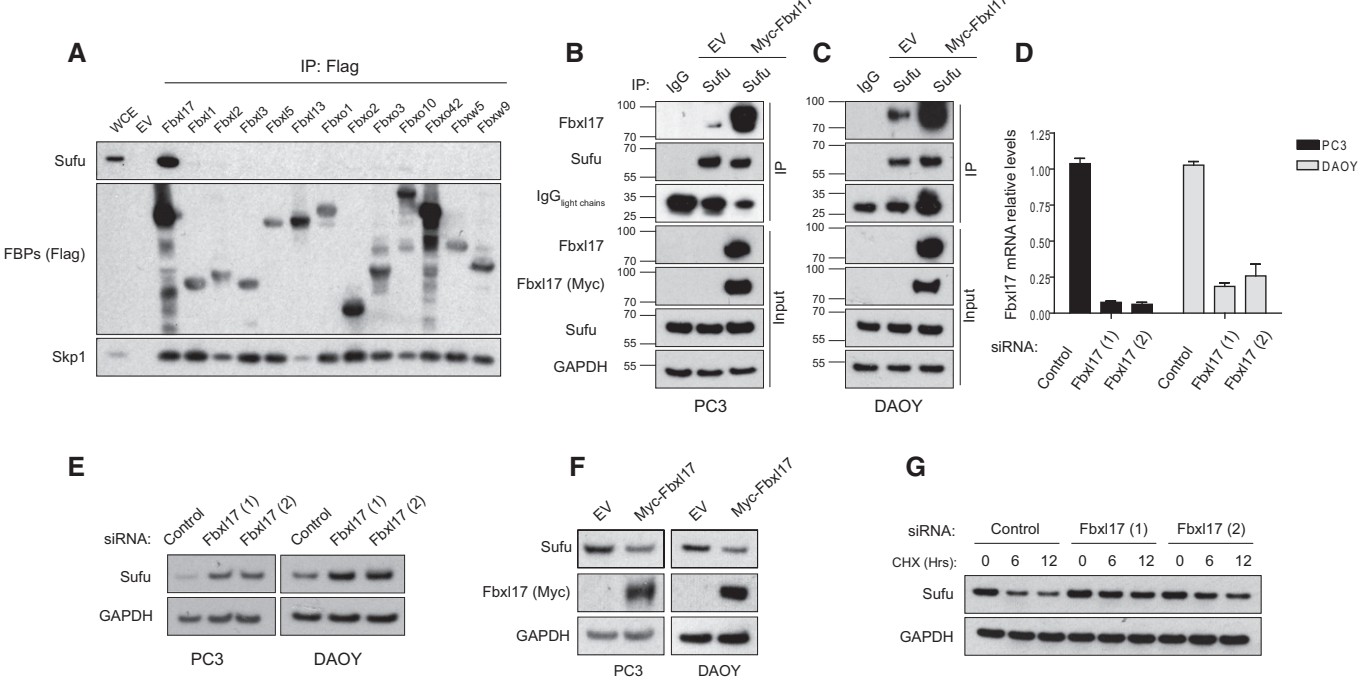

**Figure 1. Fbxl17 interacts with Sufu and regulates Sufu levels.**

A     Detection of Sufu and Skp1 after immunoprecipitation of the indicated Flag-tagged F-box proteins (FBPs). An empty vector (EV) was used as a negative control. HEK293T cells were treated with MLN4924 (2 μM) for 5 h prior to collection. Representative image of three independent experiments is shown.

B, C     Immunoprecipitation of endogenous Sufu from PC3 (B) and DAOY cells (C) transfected with either an empty vector (EV) or Myc-tagged Fbxl17. Nonspecific rabbit immunoglobulin G (IgG) was used as a negative control. Detection of light chains of immunoglobulin (IgG) was used to assess the amount of IgG used for each immunoprecipitation reaction. Treatment with MLN4925 (2 μM) was started 5 h before the cell collection.

D     Quantification of Fbxl17 mRNA levels in PC3 and DAOY cells transfected with a nontargeting siRNA (Control) or two siRNAs to Fbxl17 (1) and (2). The cells were serum-starved for 24 h in serum-reduced medium and treated with SAG (100 nM) for 24 h prior to collection. Data are shown as mean ± SEM.

E     Sufu protein levels in PC3 and DAOY cells treated as in (D). GAPDH was used as a loading control. Representative image of three independent experiments is shown.

F     Sufu protein levels in PC3 and DAOY cells infected with an empty backbone retrovirus (EV) or a retrovirus expressing Myc-tagged Fbxl17. Representative image from three independent experiments is shown.

G     Detection of Sufu protein levels following cycloheximide (CHX) treatment for the indicated hours, and Fbxl17 depletion using two different siRNAs (Hrs = h). Representative image of two independent experiments is shown.

To confirm that the binding between Fbxl17 and Sufu was not due to spurious interaction due to Fbxl17 overexpression, we isolated endogenous Sufu after treating cells with an inhibitor of NAE (Nedd8-activating enzyme) (MLN4924) (Soucy *et al*, 2009). This treatment blocks cullin neddylation required for the activity of SCF ubiquitin ligases (Ohh *et al*, 2002). Using this approach, we detected endogenous Fbxl17 in Sufu immunoprecipitates (Fig 1B and C, lane 2) using an antibody against the endogenous protein. To confirm the validity of the antibody for immunoprecipitation and Western blot, we compared Fbxl17 detected in Sufu-immunoprecipitated material to an Fbxl17, which was exogenously expressed (Fig 1B and C, lane 3). The interaction between endogenous Fbxl17 and Sufu was validated in PC3 (human prostate cancer cell line) and DAOY (medulloblastoma cancer cell line) (Fig 1B and C), indicating that Fbxl17 and Sufu interact specifically, physiologically, and in different cell lines.

To assess whether Fbxl17 affects Sufu levels, we measured the protein levels of Sufu in PC3 and DAOY cells upon Fbxl17 depletion using short interfering RNA (siRNA) (Fig 1D). Fbxl17 depletion by two siRNAs increased Sufu protein levels in both PC3 and DAOY (Fig 1E). Opposite to this, exogenous expression of Myc-tagged

Fbxl17 led to the downregulation of Sufu protein in both cell lines (Fig 1F). Sufu protein levels upon either Fbxl17 depletion or overexpression were quantified (Fig EV1A and B). Sufu mRNA levels remained unchanged upon the modulation of Fbxl17 levels (Fig EV1C and D), suggesting that Sufu is regulated by Fbxl17 through a post-translational mechanism. To exclude other mechanisms of control on Sufu by Fbxl17, we measured Sufu half-life in cells where Fbxl17 was depleted using two different siRNAs. While Sufu half-life was approximately 6 h in cells treated with a nontargeting siRNA, this increased to more than 12 h in Fbxl17-depleted cells (Fig 1G). Protein levels of Sufu, upon CHX treatment, were quantified in Fig EV1E and mRNA levels of Fbxl17 were measured by qPCR in Fig EV1F.

### SCF^Fbxl17 mediates Sufu polyubiquitylation

In order to assess whether Fbxl17 is directly targeting Sufu, we measured Sufu polyubiquitylation in cells where Fbxl17 was depleted by siRNA. Remarkably, Sufu polyubiquitylation was completely abolished by Fbxl17 siRNA in both DAOY and 293T cells (Fig 2A–C), supporting the role of Fbxl17 as the major

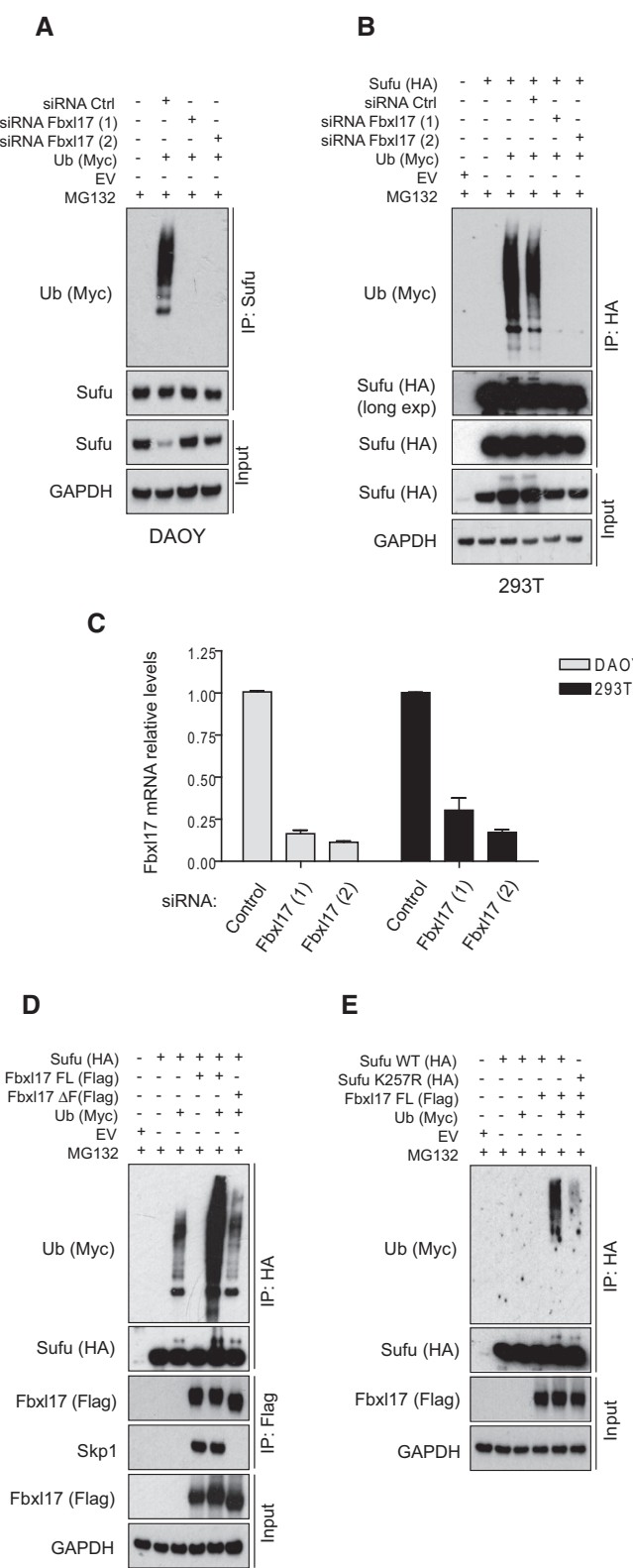

**Figure 2. SCF^Fbxl17 mediates Sufu polyubiquitylation.**

A   Detection of polyubiquitylated species of endogenous Sufu co-immunopurified from DAOY cells transfected with Myc-tagged ubiquitin (Ub) either in the presence of nontargeting siRNA (Control) or in the presence of two siRNAs against Fbxl17 (1) and (2). MG132 (10 μM) was added in all samples.

B   Detection of polyubiquitylated species of Sufu upon co-transfection of HEK293T cells with HA-tagged Sufu and Myc-tagged ubiquitin (Ub) either in the presence of nontargeting siRNA (Control) or in the presence of two siRNAs against Fbxl17 (1) and (2). MG132 (10 μM) was added in all samples.

C   Fbxl17 mRNA relative levels in DAOY and HEK293T cells upon Fbxl17 depletion with either a nontargeting siRNA (Control) or two siRNAs to Fbxl17 (1) and (2). Data are shown as mean ± SEM.

D   Detection of ubiquitylated Sufu co-immunoprecipitated from HEK293T cells co-transfected with HA-tagged Sufu, Myc-ubiquitin (Ub) along with either Flag-tagged Fbxl17 wild type (WT) or a mutant lacking the F-box domain (Fbxl17ΔF). MG132 (10 μM) was added in all samples.

E   Detection of ubiquitylated Sufu co-immunoprecipitated from HEK293T cells co-transfected with Myc-tagged ubiquitin (Ub), Flag-tagged Fbxl17, and HA-tagged Sufu WT or Sufu mutant K257R. MG132 (10 μM) was added in all samples.

binding, whereas the amino terminus and the F-box motif of Fbxl17 were dispensable for Sufu recruitment (Fig EV2A and B). To confirm that polyubiquitylation of Sufu was due to Fbxl17 activity, we measured Sufu polyubiquitylation using an Fbxl17 lacking the F-box domain (that is still able to interact with Sufu, but not with Skp1; Fig EV2B). Expression of exogenous Fbxl17 promoted Sufu polyubiquitylation, whereas an Fbxl17 mutant lacking the F-box failed to do so, indicating that the formation of the SCF is not a prerequisite for the binding of Fbxl17 to its substrates, but required for ubiquitylation (Fig 2D).

Furthermore, we observed that Fbxl17 used the lysine 257 of Sufu as a preferential attachment site for ubiquitin, since the polyubiquitylation of Sufu mutant K257R was impaired (Fig 2E). Using reticulocyte lysates, we reconstituted the components of the SCF^Fbxl17 and Sufu in an *in vitro* reaction, as previously done (Peschiaroli *et al*, 2006). We observed ubiquitylation of Sufu wild type (WT), while the Sufu K257R was ubiquitylated to a reduced extent (Fig EV2C). These findings explain the increased half-life of Sufu K257R mutant reported elsewhere (Yue *et al*, 2009) and suggest that F-box proteins might have preferential ubiquitin attachment sites on substrates.

## Fbxl17–Sufu interaction is impaired by Sufu phosphorylation and favored by the presence of Gli

To gain insights into the details of Fbxl17 recognition of Sufu, we used Sufu-truncated mutants and measured binding to Fbxl17. A region of Sufu encompassing amino acids 350–425 was found to be sufficient for the binding to Fbxl17 (Fig EV3A–C). Phosphorylation is a common mechanism used by F-box proteins for substrate recognition; however, in the case of Fbxls, phosphorylation could also block substrate recognition (Kuchay *et al*, 2013; Skaar *et al*, 2013).

Sufu WT was unable to bind Fbxl17 efficiently as the Sufu 350–484-truncated mutant (Fig EV3B), suggesting the presence of a phosphorylation event on Sufu, that might prevent Fbxl17 binding outside of the minimal binding region. Thus, we systematically mutagenized all the serine and threonine residues to alanine on Sufu between the residues 1 and 350. While the majority of sites did

physiological ubiquitin ligase for Sufu in different cellular systems.

Deletion mapping analysis showed that the carboxyl terminus of Fbxl17 containing the leucine zipper motif is essential for Sufu

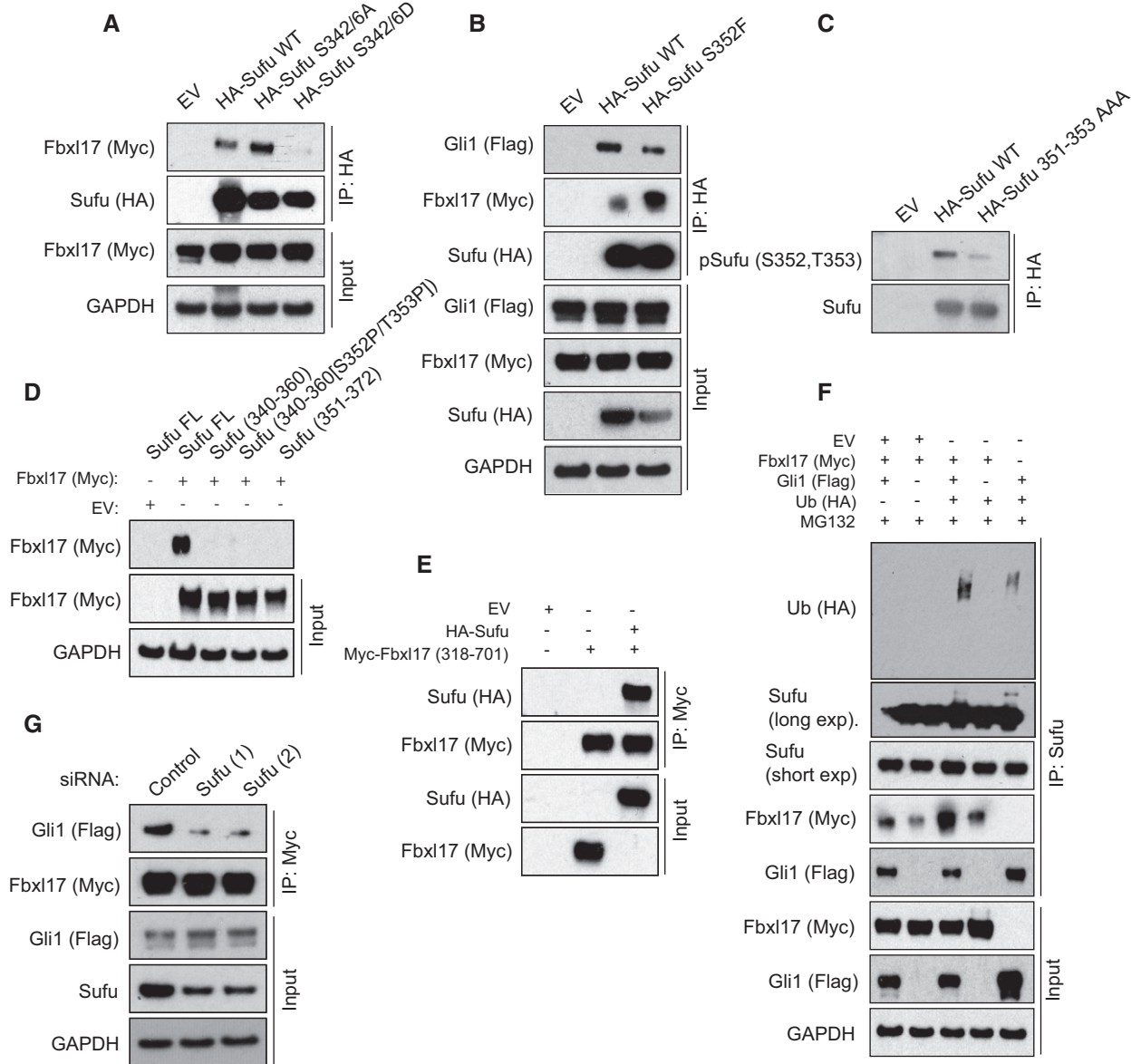

**Figure 3.  Fbxl17–Sufu interaction is impaired by Sufu phosphorylation and favored by the presence of Gli.**

A   Detection of Myc-tagged Fbxl17 after immunoprecipitation of HA-tagged Sufu WT or Sufu S342/6A and S342/6D, as indicated. HEK293T cells were treated with MLN4924 (2 μM) for 5 h prior to collection.

B   Detection of Flag-tagged Gli1 and Myc-tagged Fbxl17 binding to immunoprecipitated HA-tagged Sufu wild type (WT) or to Sufu S352F. An empty vector (EV) was used as a negative control. HEK293T cells were treated with MLN4924 (2 μM) for 5 h prior to collection.

C   Detection of Sufu phosphorylated on S352/T353 after immunoprecipitation of HA-tagged Sufu WT and Sufu 351–353 AAA, as indicated.

D   Detection of Myc-tagged Fbxl17 binding to Sufu full-length (FL) or to the following Sufu peptides: Sufu 340–360 (APSRKDSLESDSSTAIIPHEL); Sufu 340–360 [S352P/ T353P] (phosphorylated on the residues S352 and T353); Sufu 351–372 (SSTAIIPHELIRTRQLESVHLK).

E   Detection of HA-tagged Sufu binding to a Myc-tagged Fbxl17 construct encompassing the residues 318–701 (corresponding to the F-box motif and C-terminus region) assessed by *in vitro* binding assay. Both proteins were synthesized *in vitro* using a T7-coupled reticulocyte lysate system.

F   Detection of ubiquitylated Sufu immunoprecipitated from HEK293T cells co-transfected with HA-tagged ubiquitin (Ub), Myc-tagged Fbxl17, and Flag-tagged Gli1. MG132 (10 μM) was added in all samples.

G   Detection of Flag-tagged Gli1 binding to Myc-tagged Fbxl17 co-immunoprecipitated from HEK293T cells transfected with a nontargeting siRNA (Control) or two siRNA to Sufu (1) and (2), as indicated.

not affect the binding of Fbxl17 to Sufu (Fig EV3E and F), a mutant of Sufu in which the S342 and S346 residues were changed to alanine (A) increased binding to Fbxl17 (Fig 3A). Substitution of S342 and S346 to D (S342/6D) to mimic phosphorylation decreased

recognition by Fbxl17 (Fig 3A). These data are consistent with the increased stabilization of Sufu detected upon the phosphorylation of these residues by PKA (protein kinase A) and GSK3β (glycogen synthase kinase 3β) (Chen *et al*, 2011).

Within the minimal binding region of Sufu to Fbxl17, Sufu has been found phosphorylated on S352 (Hsu *et al*, 2011). A natural occurring mutation (S352F) abolishing S352 phosphorylation has been described in the medulloblastoma of patients affected by Gorlin syndrome (Smith *et al*, 2014). Therefore, we tested Fbxl17 binding to Sufu S352F. The Sufu S352F substitution increased Fbxl17 binding (Fig 3B); on the contrary, a mutant of Sufu where the S352 residue was changed to aspartate (D) to mimic the phosphorylation showed reduced binding to Fbxl17 (Fig EV3D). We performed LC/MS analysis to obtain high coverage of Sufu modifications and identified multiple phosphorylation sites in this region (Appendix Fig S1A). Furthermore, we generated a phospho-specific antibody against a peptide of Sufu containing both S352 and T353 residues phosphorylated. The antibody recognizes Sufu WT, indicating that S352 and T353 are phosphorylated *in vivo*, but did not recognize a mutant of Sufu where S352 and T353 were substituted to alanine (S351-T353AAA) (Fig 3C).

Given the crucial role of residues S342, S346, and S352 for Fbxl17–Sufu interaction, we tested whether Fbxl17 is able to bind to a linear peptide encompassing amino acids 340–360 (with and without the phosphorylation of S352/T353 residues) and 351–372. Independently of the phosphorylation status of S352 residue, Fbxl17 purified from the cells did not bind to Sufu peptides, but interacted with full-length Sufu *in vitro*, indicating that a linear sequence is not sufficient for Fbxl17 binding (Fig 3D). Additionally, Fbxl17 and Sufu synthesized *in vitro* were able to bind, pinpointing that the interaction is direct and independent of post-translational modifications induced by Hh signaling (Fig 3E).

Fbxws utilize the WD40 domain to interact with phosphorylated linear degradation sequences; however in the case of the Fbxls, the leucine zipper motif is contacting a larger surface and could require the presence of cofactors (Hao *et al*, 2005; Xing *et al*, 2013). The conformation of Sufu is regulated by Gli binding and Sufu–Gli interaction requires an intact tertiary structure of Sufu (Cherry *et al*, 2013; Zhang *et al*, 2013). Thus, we asked whether Gli1 acts as a cofactor modulating the interaction between Fbxl17 and Sufu *in vivo*. Remarkably, Fbxl17 polyubiquitylation of endogenous Sufu was favored by the presence of exogenous Gli1 that promoted an increased binding of Fbxl17 to Sufu and the subsequent polyubiquitylation (Fig 3F). To assess that the latter was due to the direct interaction between Fbxl17 and Sufu, we depleted Sufu using two different siRNA and tested the binding of Gli1 to Fbxl17. Sufu depletion abolished the binding of Gli1 to Fbxl17, indicating that Sufu bridges the interaction between Fbxl17 and Gli1 *in vivo* (Fig 3G).

## Sufu ubiquitylation by Fbxl17 allows Gli dissociation for Hedgehog signaling activation

It is not well understood how Sufu is released from Gli1 for full Hh pathway activation. We speculated that Fbxl17 could mediate the release by promoting Sufu polyubiquitylation and degradation. To assess the latter, we analyzed Sufu protein levels in mouse embryonic fibroblasts (MEFs) that express the receptor Ptch1 ($Ptch1^{+/+}$), MEFs with heterozygous loss of $Ptch1$ ($Ptch1^{+/-}$), and MEFs deficient in $Ptch1$ ($Ptch1^{-/-}$), which have constitutive Hh signaling (Taipale *et al*, 2000). Strikingly, MEFs $Ptch1^{-/-}$ and $Ptch1^{+/-}$ cells showed a decrease in Sufu levels corresponding to the degree of Ptch loss and the consequent Hh activation. Sufu levels were reduced in $Ptch1^{-/-}$ and less significantly in $Ptch1^{+/-}$, while remained unaffected in $Ptch1^{+/+}$, indicating that Fbxl17 is acting downstream of Ptch and Smo. Low levels of Sufu present in MEFs $Ptch1^{+/-}$ and MEFs $Ptch1^{-/-}$ were restored to levels comparable to $Ptch1^{+/+}$ by Fbxl17 depletion (Fig 4A and B). Furthermore, MEFs $Ptch1^{-/-}$ lacking Fbxl17 presented low transcription levels of Hh target genes such as Gli1 and Bcl2 due to a negative effect that Sufu accumulation exerts on Gli-mediated target gene expression (Fig 4C and D).

Considering the defects observed in pathway activation upon Fbxl17 depletion in MEFs $Ptch1^{-/-}$ cells and that Fbxl17 preferentially binds to the Sufu–Gli complex, we speculated that Fbxl17 polyubiquitylation favors Sufu–Gli complex dissociation to allow the need to be substituted with Hh pathway activation. Indeed, Fbxl17 overexpression induced Sufu degradation and Gli1 release, whereas the depletion of Fbxl17 in MEFs $Ptch1^{-/-}$ led to an accumulation of Sufu–Gli1 complexes (Fig 4E and Appendix Fig S1B). Since the polyubiquitylation of Sufu precedes degradation, this model is in accordance with previous findings on Sufu–Gli complex formation and dissociation (Tukachinsky *et al*, 2010).

Finally, we addressed where the degradation of Sufu takes place in cells MEFs $Ptch1^{-/-}$ where the pathway is constitutively activated. Due to the lack of an antibody detecting endogenous Fbxl17 by immunofluorescence, we stably expressed Fbxl17 using a retroviral system that maintains physiological levels of protein expression. To avoid localization artifacts, we expressed an Fbxl17 tagged with

**Figure 4. Sufu ubiquitylation by Fbxl17 allows Gli dissociation for Hedgehog signaling activation.**

A   Sufu protein levels in mouse embryonic fibroblasts (MEFs) $Ptch1^{+/+}$, $Ptch1^{+/-}$, and $Ptch1^{-/-}$ transfected with a nontargeting siRNA (Control) or two siRNAs targeting mouse Fbxl17 (1) and (2).

B   Quantification of Fbxl17 mRNA levels in MEFs $Ptch1^{+/+}$, $Ptch1^{+/-}$, and $Ptch1^{-/-}$ transfected as in (A). Data are shown as mean ± SEM.

C   Analysis of Gli1 mRNA levels in MEFs $Ptch1^{+/+}$ and $Ptch1^{-/-}$ upon Fbxl17 depletion using two different siRNAs (mean ± SEM from three independent experiments, ***$P < 0.0005$, unpaired *t*-test).

D   Analysis of Bcl2 mRNA levels in MEFs $Ptch1^{+/+}$ and $Ptch1^{-/-}$ upon Fbxl17 depletion using two different siRNAs (mean ± SEM from three independent experiments, **$P < 0.005$, ***$P < 0.0005$, unpaired *t*-test).

E   Detection of Flag-tagged Gli1 binding to HA-tagged Sufu in MEFs $Ptch1^{-/-}$ upon expression of Myc-tagged Fbxl17, or Fbxl17 depletion by two siRNAs. For each condition, 200 μg of whole-cell lysate was immunoprecipitated.

F   Immunostaining of Fbxl17 using anti-Myc antibody in $Ptch1^{-/-}$ MEFs stably transduced with a pBABE vector expressing Fbxl17 tagged with Myc either at the N-terminus [Myc(N)-Fbxl17] or at the C-terminus [Myc(C)-Fbxl17]. Scale bars: 20 μm.

G   Detection of Fbxl17 using anti-Myc antibody in $Ptch1^{-/-}$ MEFs stably expressing Myc(N)-Fbxl17 (Fbxl17 tagged at the N-terminus) and Myc(C)-Fbxl17 (Fbxl17 tagged at the C-terminus).

H   Detection of HA-tagged Fbxl17 binding to endogenous Sufu immunoprecipitated from cytoplasmic and nuclear extracts from $Ptch1^{-/-}$ MEFs. Asterisk (*) indicates a nonspecific band. Identification of cytoplasmic and nuclear fractions was performed by lamin A/C and GAPDH detection.

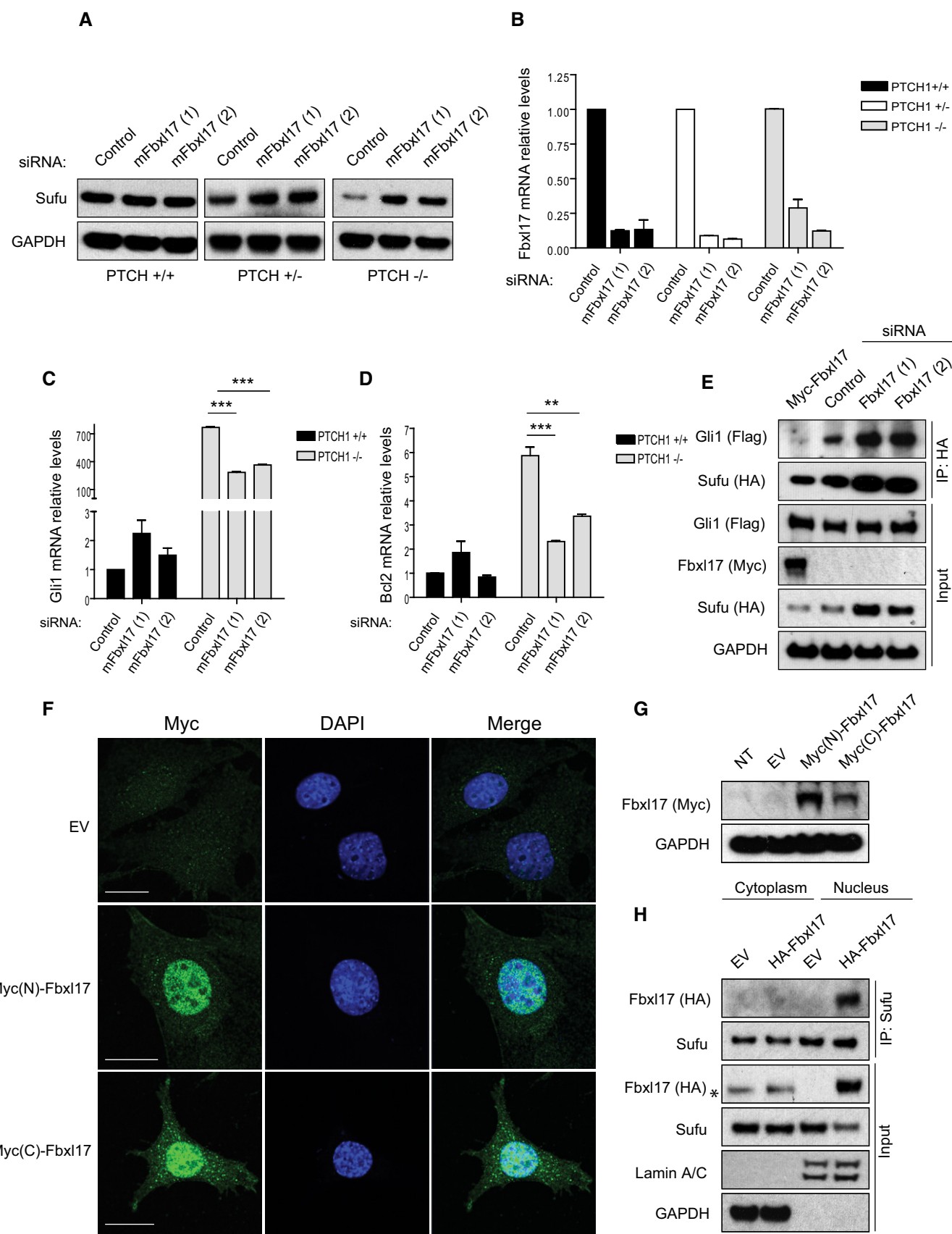

**Figure 4.**

Myc at the C-terminus and N-terminus. In both cases, the localization of Fbxl17 in MEFs *Ptch1*$^{-/-}$ was exclusively nuclear (Fig 4F and G). Furthermore, exogenous Fbxl17 expressed in MEFs *Ptch1*$^{-/-}$ induced downregulation of Sufu in the nucleus, where Fbxl17 interacted with Sufu upon biochemical fractionation of the cytoplasmic and nuclear extracts (Fig 4H). Since phosphorylation of Sufu on S342/S346 promotes ciliary retention of Sufu (Chen *et al*, 2011), these data are in accordance with a model where Sufu dephosphorylation promotes the localization outside the cilium and ubiquitin mediated proteolysis by Fbxl17.

### Fbxl17-mediated degradation of Sufu regulates Hh signaling, cancer cell proliferation, and medulloblastoma growth

Given the relevant role of Fbxl17 in regulating Hh signaling and that Hh signaling activity has been implicated in the proliferation of prostate cancer cell lines (Karhadkar *et al*, 2004; Sanchez *et al*, 2004), we further tested the effect of Fbxl17 on Hh activation by quantifying Gli1 mRNA levels upon either depletion or expression of Fbxl17 using PC3 cells. Fbxl17 silencing resulted in a decrease in Gli1 mRNA levels (Appendix Fig S2A), and contrarily, Fbxl17 expression led to an increase in Gli1 mRNA levels (Appendix Fig S2B). Since proliferation of PC3 cells is in part dependent on Hh signaling pathway (Karhadkar *et al*, 2004; Zhang *et al*, 2007), we evaluated whether Fbxl17 depletion impairs the prostate cancer cell proliferation. Fbxl17 silencing induced a defective cell proliferation of PC3 cells and this defect was rescued by the reintroduction of Fbxl17 full length in cells depleted of Fbxl17 (Appendix Fig S2C–F). Moreover, this phenotype was dependent on the accumulation of Sufu, as double siRNA of Fbxl17 and Sufu could restore the proliferation (Appendix Fig S3A–D).

A more direct role of Hh pathway has been established in the pathogenesis of medulloblastoma characterized by the presence of mutations in central components of Hh signaling (Kool *et al*, 2014); therefore, we tested whether the Fbxl17–Sufu axis operates in medulloblastoma cancer cells. DAOY human medulloblastoma cells have been reported to be responsive to Hh signaling modulation (Gotschel *et al*, 2013). DAOY cells responded to Smo activation (using SAG; Smo agonist) or inhibition (using cyclopamine; Smo antagonist) by upregulating or downregulating Gli1 mRNA levels, respectively (Fig EV4A). Silencing of Fbxl17 impaired Gli1 transcription upon SAG-induced Hh activation (Fig EV4B–D). Additionally, upon the stimulation of DAOY cells with SHH (Sonic Hedgehog)-conditioned media (Fig EV4E), Sufu protein levels were reduced (Fig EV4F). Fbxl17 depletion restored Sufu protein levels after SHH-conditioned media (Fig EV4F and quantified in EV4G), confirming that the downstream activation of Hh signaling pathway depends on Fbxl17-mediated degradation of Sufu in DAOY cells. The observed downregulation of Sufu protein was not due to the variation in Fbxl17 expression upon Hh stimulation. Indeed, the mRNA of Fbxl17 was not affected in DAOY cells upon Hh stimulation with SAG or SHH ligand and Hh inhibition with cyclopamine (Fig EV4H and I).

Similar to PC3 cells, we evaluated whether Fbxl17 depletion impairs DAOY cell proliferation. Proliferation of DAOY cells was hampered by Fbxl17 depletion and stimulated by Fbxl17 re-expression (Fig 5A). Fbxl17 siRNA (quantified in Fig 5B) increased protein levels of Sufu; on the contrary, expression of Fbxl17 corresponded to low protein levels of Sufu in DAOY (Fig 5C and Appendix Fig

S4A). Importantly, the proliferation defect induced by Fbxl17 depletion was dependent on the accumulation of Sufu, as double siRNA of Fbxl17 and Sufu could restore proliferation in DAOY (Fig 5D–F and Appendix Fig S4B).

To investigate the relevance of Fbxl17–Sufu axis in cancer, we assessed the effect of RNAi of Fbxl17 *in vivo* by using an orthotopic rat model of medulloblastoma. Fbxl17 RNAi led to an impaired Hh signaling reflected by the accumulation of Sufu protein and decreased Gli1 mRNA (Fig 6A–C and Appendix Fig S5A).

*In vivo* $T_2$-weighted magnetic resonance imaging (MRI) showed a marked reduction in tumor progression over time with RNAi against Fbxl17 (Fig 6D and E). In contrast, the area of gadolinium enhancement on $T_1$-weighted images at the end-point showed no differences between groups (Appendix Fig S5B and C). Histologically, tumors with RNAi against Fbxl17 showed significantly reduced growth compared to a nontargeting control as detected by vimentin staining ($6.1 \pm 1.7$ mm$^2$ vs. $12.4 \pm 1.3$ mm$^2$; $P < 0.05$; Fig 6F and G), and also a significantly lower proliferative index within the tumor tissue ($5.4 \pm 3.6\%$ vs. $20.8 \pm 2.8\%$; $P < 0.05$; Fig 5H and I). Together, these data suggest that $T_2$-weighted MRI may be more sensitive to inhibition of tumor growth than post-contrast $T_1$-weighted MRI in this setting.

### Fbxl17–Sufu axis is altered in medulloblastoma

According to the gene transcription profiles, four different subtypes of medulloblastoma have been described, that is, wingless type (Wnt), Sonic Hedgehog (SHH), group 3, and group 4 (Gajjar & Robinson, 2014). The SHH subtype of medulloblastoma is characterized by an increased Hh signaling due to the altered expression of pathway components. Given the relevant role of Fbxl17 in Hh signaling and medulloblastoma growth, we analyzed the mRNA expression of Fbxl17 and Sufu reported in the largest published medulloblastoma expression profiling study ($N = 285$ primary samples) (Northcott *et al*, 2012). Although Sufu has a pivotal role in Hh signaling and mutations in Sufu are present in SHH medulloblastoma, the mRNA of Sufu was found to be not reduced in the SHH subtype, whereas Gli1 was found to be drastically increased (Fig 7A and B). Strikingly, Fbxl17 mRNA was found to be significantly increased in the SHH-subtype medulloblastoma (Fig 7C) and the levels of Fbxl17 and Gli1 positively correlated (Spearman's rho = 0.5640; $P < 0.00001$; Fig 7D). In the absence of Sufu mRNA alterations, these results suggest that in medulloblastoma the control of Sufu protein levels by Fbxl17 plays a prominent role in sustaining Hh activation.

The distinction of four molecular subgroups depends on the transcriptional profiles of the subgroups, which at least for SHH subtype reflects the cells of origin of the medulloblastoma. In the SHH subtype is established that granule cell progenitors (GCPs) represent the cell of origin of the malignancy (Wechsler-Reya & Scott, 1999; Marino *et al*, 2000; Oliver *et al*, 2005; Schuller *et al*, 2008). Thus, we tested the effect of siRNA and the expression of Fbxl17 in GCPs. Similar to the results obtained in DAOY, Fbxl17 expression increased cell proliferation (Fig EV5A and B); on the contrary, siRNA of Fbxl17 reduced cell proliferation measured by BrdU incorporation (Fig EV5C and D).

To establish a role for Fbxl17 in the etiology of medulloblastoma through Sufu mutation, we investigated in more detail the Sufu

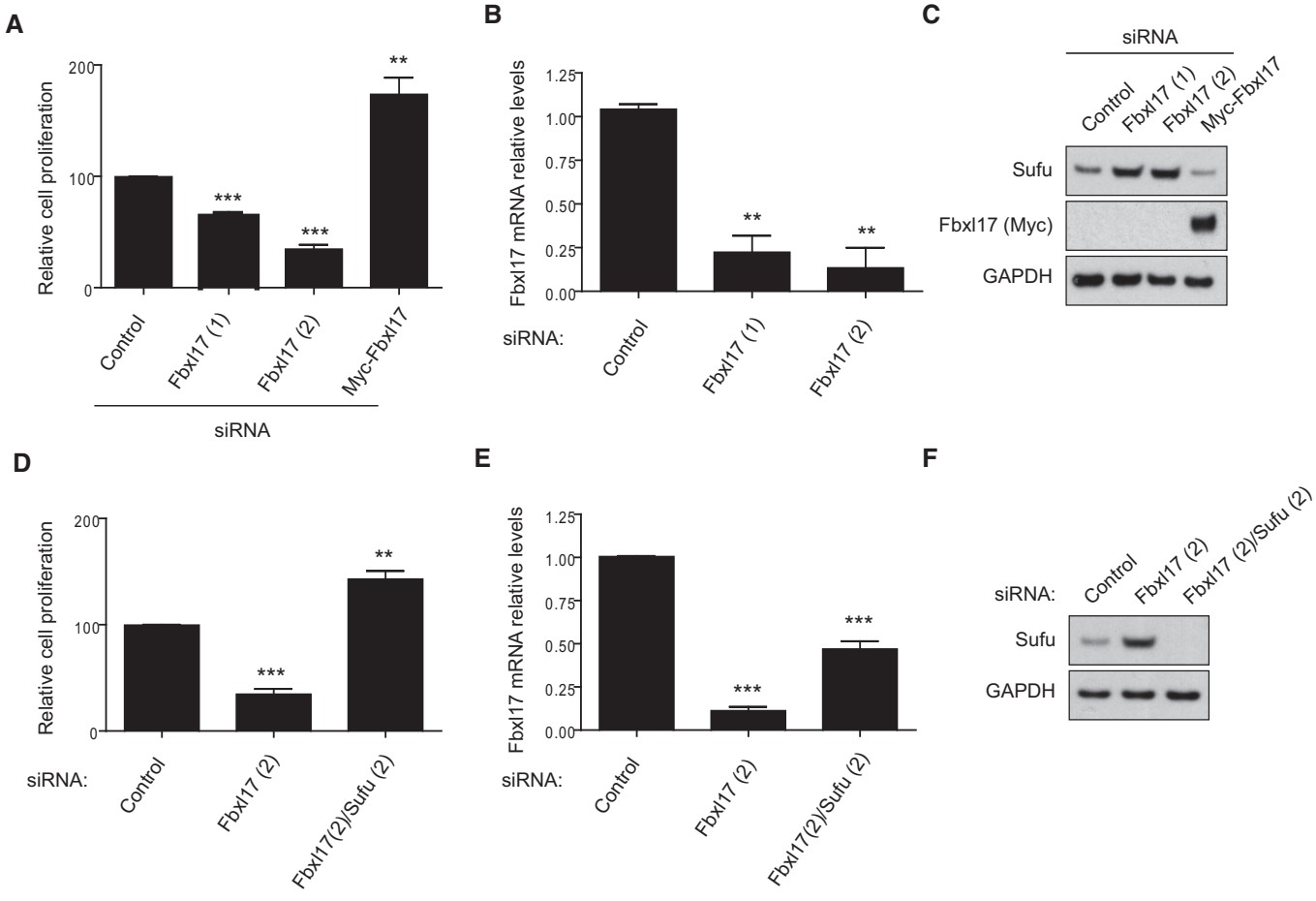

**Figure 5. Fbxl17-mediated degradation of Sufu promotes medulloblastoma cell proliferation.**

A   Cell proliferation of DAOY cells upon Fbxl17 depletion using a nontargeting siRNA (Control), two siRNAs against Fbxl17 (1) and (2) or upon reintroduction of a Myc-tagged Fbxl17 construct in Fbxl17-depleted DAOY cells using siRNA Fbxl17 (2) (mean ± SEM from three independent experiments, **$P < 0.005$; ***$P < 0.0005$, unpaired *t*-test).

B   Quantification of Fbxl17 mRNA levels in DAOY cells transfected with a nontargeting siRNA (Control) or two siRNAs against Fbxl17 (1) and (2) (mean ± SEM from three independent experiments, **$P < 0.005$, unpaired *t*-test).

C   Sufu protein levels in DAOY cells treated as in (A). Representative image of three independent experiments is shown.

D   Cell proliferation of DAOY cells transfected with nontargeting siRNA (Control), siRNA against Fbxl17 or a combination of siRNA targeting Fbxl17 and Sufu (mean ± SEM from three independent experiments, **$P < 0.005$; ***$P < 0.0005$, unpaired *t*-test).

E   Quantification of Fbxl17 mRNA levels in DAOY cells treated as in (D) (mean ± SEM from three independent experiments, ***$P < 0.0005$, unpaired *t*-test).

F   Sufu protein levels in DAOY cells treated as in (D). Representative image of three independent experiments is shown.

Source data are available online for this figure.

S352F substitution, which favors binding to Fbxl17 (Fig 3A). This substitution appears in the medulloblastoma of patients affected by Gorlin syndrome and contributes to the medulloblastoma development, in the absence of other alterations in Hh pathway (Smith *et al*, 2014). We observed that the half-life of Sufu S352F was found to be decreased compared to Sufu WT, in accordance with an increased binding to Fbxl17. Furthermore, silencing of Fbxl17 restored Sufu mutant half-life (Fig 7E). Using firefly luciferase-based Gli-reporter assay (Sasaki *et al*, 1997), we assessed the effect of Sufu S352F on Hh activation in MEFs *Sufu*$^{-/-}$ reconstituted with either Sufu WT or SufuS352F. Strikingly, Sufu S352F increased the responsiveness of cells to Hh agonist (SAG) (Fig 7F), indicating that the faster proteolysis of Sufu, mediated by Fbxl17, has a functional role in the control of Hh signaling.

These observations exemplify the role of Fbxl17–Sufu axis alterations in medulloblastoma.

## Discussion

Our findings identify Fbxl17 as a novel regulator of Hh signaling pathway. Fbxl17 mediates Sufu polyubiquitylation and degradation through the proteasome. The interaction between Fbxl17 and Sufu is hampered by the phosphorylation of Sufu on S342 and S346 residues, which were shown to be target sites for PKA and GSK3β, respectively (Chen *et al*, 2011) (Fig 8A). Phosphorylation of Sufu retains it in the primary cilium (Chen *et al*, 2011) and could represent a failsafe mechanism to avoid unscheduled pathway activation

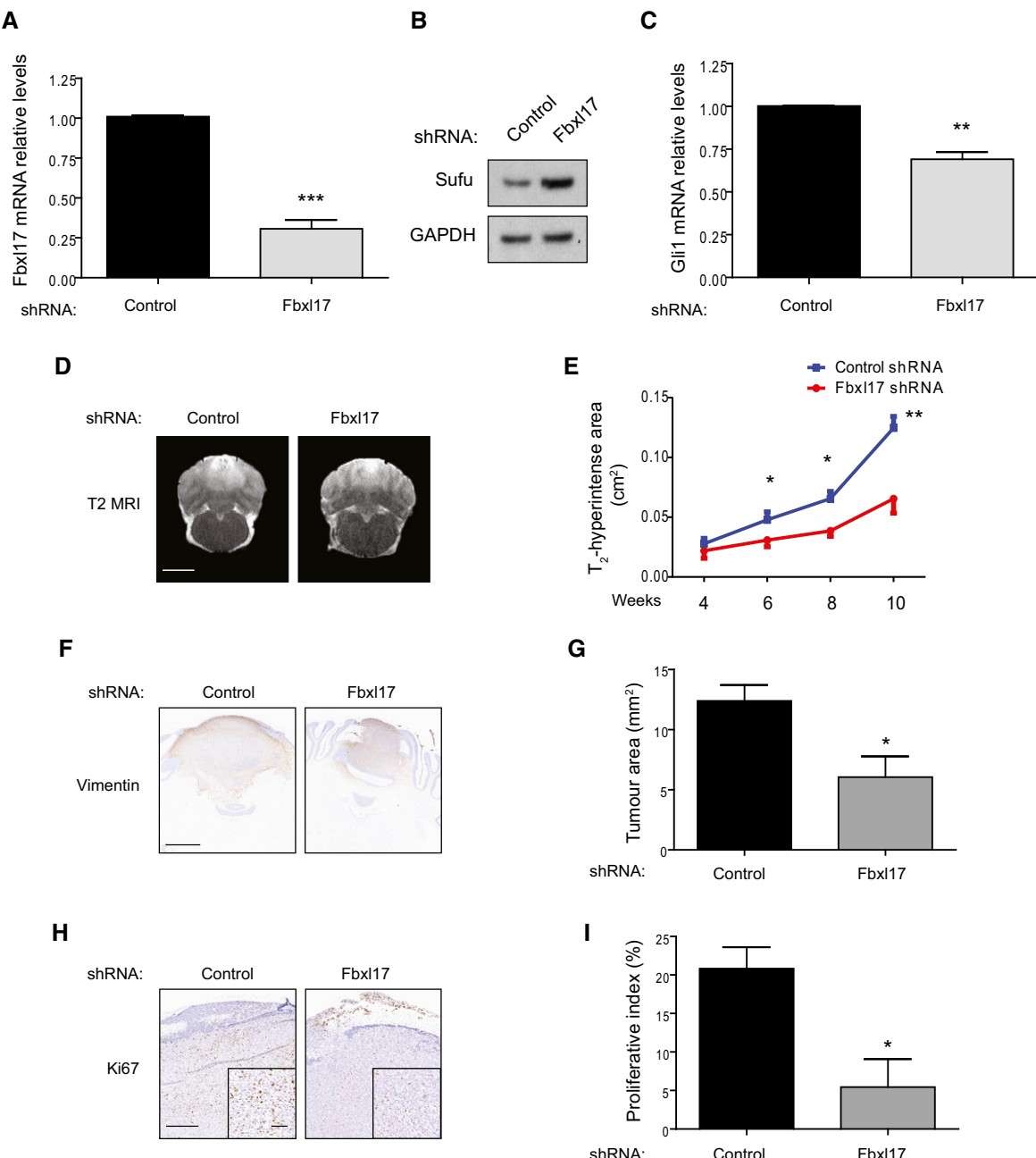

**Figure 6.  Fbxl17-mediated degradation of Sufu promotes medulloblastoma tumor growth.**

A    Quantification of Fbxl17 mRNA levels in DAOY cells transfected with control shRNA or shRNA targeting Fbxl17 (mean ± SEM from three independent experiments, ***$P < 0.0005$, unpaired *t*-test).

B    Detection of Sufu protein levels in DAOY cells transfected with control shRNA or shRNA Fbxl17. Representative image of three independent experiments is shown.

C    Quantification of Gli1 mRNA levels in DAOY cells transfected with control shRNA or shRNA Fbxl17 (mean ± SEM from three independent experiments, **$P < 0.005$, unpaired *t*-test).

D    $T_2$-weighted magnetic resonance images showing tumor development in rats injected with DAOY cells transfected with control shRNA or shRNA against Fbxl17. One representative image for each condition at 10 weeks is shown. Scale bar: 5 mm.

E    Graph showing area of $T_2$ hyperintensity (a surrogate marker of tumor growth) between 4 and 10 weeks post-tumor induction. Animals were injected with DAOY cells stably expressing control shRNA or shRNA against Fbxl17 (10,000 cells/µl) (mean ± SEM; $n = 5$; two-way ANOVA, followed by unpaired *t*-test, *$P < 0.05$, **$P < 0.01$).

F    Representative immunohistochemical image of vimentin staining. Scale bar: 1 mm.

G    Quantification of tumor growth (vimentin staining) in rats injected with DAOY cells transfected with either control shRNA or shRNA against Fbxl17 (mean ± SEM; $n = 5$; unpaired *t*-test, *$P < 0.05$).

H    Representative immunohistochemical image of Ki67 staining. Scale bar: 0.5 mm/0.1 mm.

I    Quantification of proliferative index (Ki67 staining) in rats injected with DAOY cells transfected with either control shRNA or shRNA against Fbxl17 (mean ± SEM; $n = 5$; unpaired *t*-test, *$P < 0.05$).

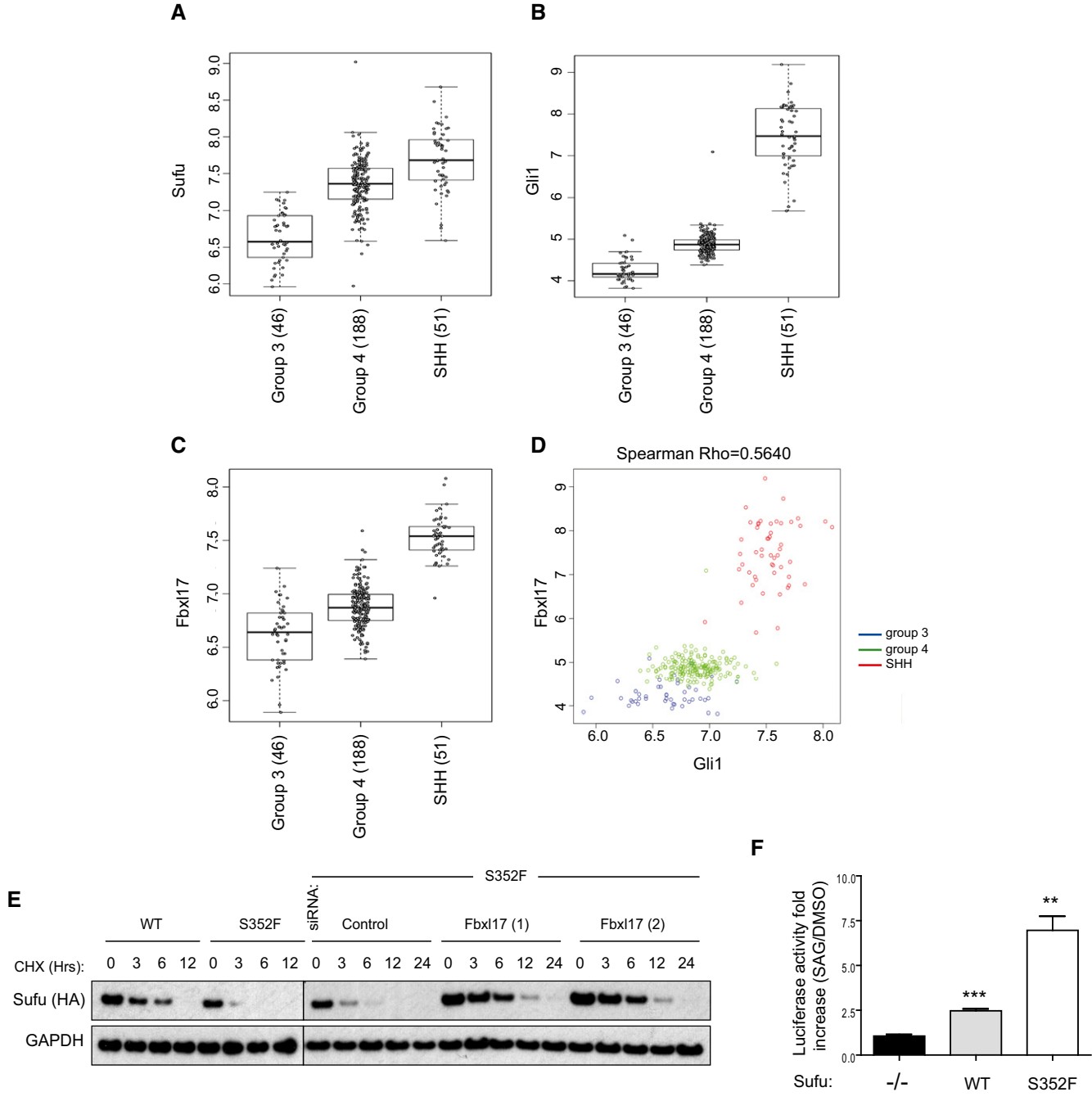

**Figure 7. Fbxl17–Sufu axis is altered in medulloblastoma.**

A–C   mRNA expression of Sufu, Gli1, and Fbxl17 detected in 285 medulloblastoma samples (GEO accession: GSE37382), stratified by tumor subtype. Level of significance (P) for Kruskal–Wallis (KW) rank test is $P < 10^{-8}$. Analysis using one-way ANOVA gave comparable results (median $\pm$ 1.5 interquartile distance).

D     Expression of Gli1 and Fbxl17 in the 285 patients is shown colored by subgroup; level of significance (P) for Kruskal–Wallis (KW) rank test is $P < 10^{-8}$, and Spearman's rank correlation (R = 0.5640) is indicated.

E     Detection of HA-tagged Sufu wild type (WT) and HA-tagged Sufu S352F protein levels in NIH3T3 cells after cycloheximide (CHX) treatment. Where indicated, the cells were transfected with a nontargeting siRNA (Control) or two siRNA to Fbxl17 (1) and (2). Representative image of three independent experiments is shown.

F     Assessment of Hh pathway activation using Gli1-luciferase reporter assay in MEFs $Sufu^{-/-}$ upon the reintroduction of HA-tagged Sufu WT or Sufu mutant S352F using a retroviral system (mean $\pm$ SEM from three independent experiments, **$P < 0.005$; ***$P < 0.0005$, unpaired $t$-test).

when the Hh ligands are not present. A similar but opposite mechanism of regulation is operated downstream of Sufu at the levels of Gli2 and Gli3 by the SCF βTrcp ubiquitin ligase. In this case, the

concerted action of PKA and GSK3β generates a phosphorylation-dependent signal for ubiquitylation by βTrcp (Bhatia *et al*, 2006; Wang & Li, 2006). The ubiquitylation of Gli2 and Gli3 generates

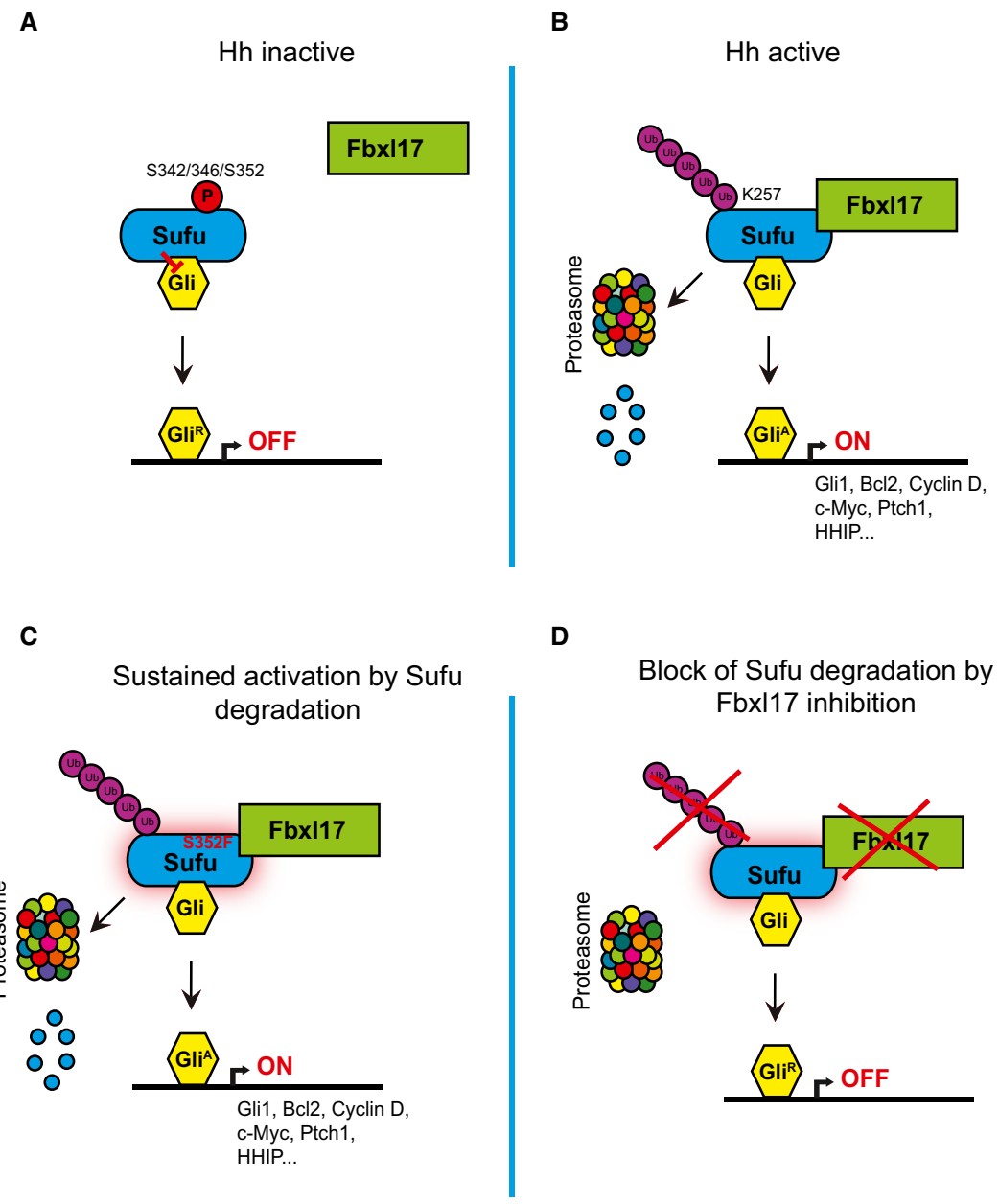

**Figure 8.  SCF (Fbxl17) ubiquitylation of Sufu regulates Hedgehog signaling and medulloblastoma growth.**

A–D   Scheme outlining the mechanism of Sufu recognition by Fbxl17 and its alteration in medulloblastoma.

transcriptional repressors to avoid pathway activation. Thus, PKA and GSK3β act on multiple components of Hh pathway as general negative regulators (Chen & Jiang, 2013).

Upon Hh ligand binding to Ptch1, a swift activation of the pathway entails the reversal of PKA and GSK3β phosphorylation on Sufu S342 and S346, in addition to S352/T353. The dephosphorylation of Sufu initiates Fbxl17 polyubiquitination of Sufu on lysine 257 for proper signal transduction (Fig 8B). Sufu is upstream of Gli processing (Humke *et al*, 2010), and thus, it is tantalizing to speculate that the dephosphorylation of Sufu promotes Gli dephosphorylation for the formation of transcriptional activators. In

medulloblastoma, increased expression of Fbxl17 and substitution in critical Sufu residues (S352F), required for Fbxl17 binding, allow amplified pathway activation and increased cell proliferation (Fig 8C). Inhibition of Fbxl17 blocks Sufu degradation, Hh pathway activation and prevents medulloblastoma tumor growth (Fig 8D).

An important feature of Fbxl17–Sufu axis is that Fbxl17 acts preferentially on Sufu in complex with Gli1, thus selecting for degradation an active pool of Sufu. Upon Hh signaling activation, Fbxl17-mediated polyubiquitylation of Sufu induces Sufu–Gli complex dissociation for the transduction of the Hh signal.

Sufu protein levels are exquisitely sensitive to the degree of Hh activation, indicating that the regulation of Sufu levels by Fbxl17 adds a further layer of control for the generation of more refined transcriptional programs in physiological conditions. Indeed, Fbxl17 could regulate Sufu levels to fine-tune transcriptional outputs; however, more studies are required to establish how the upstream components of the pathway regulate Sufu and Fbxl17. At the same time, it is noteworthy that Fbxl17 transcription is blocked by methylation in embryonic stem cells and transcribed in neural stem cells where Hh plays a more active role (Cortese *et al*, 2011). In line with this, Fbxl17 expression is increased in the brain and spinal cord during development (Diez-Roux *et al*, 2011), pointing out to a role for Fbxl17 in physiological development. Furthermore, a putative homolog of Fbxl17 (CG31633) has been described in *D. melanogaster*, where the absence of the latter induces a pupal lethal phenotype (Dui *et al*, 2012), suggesting an essential role of Fbxl17 in development.

Mutations in the components of Hh signaling are a feature of SHH medulloblastoma (Kool *et al*, 2014). We show that in a case of familial medulloblastoma, altered proteolysis of Sufu plays an active role in the induction of sustained Hh signaling. Importantly, mutations in other components of Hh signaling pathway, such as Ptch (present in > 50% of SHH medulloblastoma) (Kool *et al*, 2014), will also affect Sufu protein levels, given the effect of pathway activation on Sufu protein mediated by Fbxl17. Thus, in the majority of SHH medulloblastoma, Sufu protein levels will be low due to the constitutive activation of Hh signaling. In these cancers, increasing Sufu levels through Fbxl17 inhibition could block Hh signaling and cell proliferation. Overexpression of Fbxl17 in SHH subgroups provides a therapeutic window to selectively target cancer cells. Thus, our study highlights Fbxl17 as a novel target for the treatment of SHH medulloblastoma.

Other mutations of Sufu residing in critical residues for Fbxl17 binding have been described in colorectal cancer and malignant melanoma (Krauthammer *et al*, 2012; Giannakis *et al*, 2014); thus, alteration in Fbxl17–Sufu axis could be present in other cancer types. The biological significance of these alterations for cell proliferation remains to be determined in these cancers. The use of Smo inhibitors for the treatment of medulloblastoma is being tested in the clinic with encouraging results (Robinson *et al*, 2015), but could be significantly prevented by the acquisition of resistance through secondary Smo mutations, described in basal cell carcinoma and medulloblastoma (Yauch *et al*, 2009; Atwood *et al*, 2015; Sharpe *et al*, 2015). In medulloblastoma, the combination of Smo inhibitors and Fbxl17 inhibitors could be a viable alternative strategy to obtain sustained responses.

Our findings pave the way for the investigation of therapies targeting the ubiquitin proteasome system in the treatment for medulloblastoma and other cancers relying on Hh signaling.

# Materials and Methods

### Cell culture

All cell lines were routinely cultured in a humidified incubator at 37°C under 5% $CO_2$ in the indicated culture medium containing 10% fetal bovine serum (FBS, Sigma Aldrich) supplemented with 100 U/ml penicillin sodium and 100 μg/ml streptomycin sulfate (Sigma Aldrich).

HEK293T, HeLa (ATCC, American Type Culture Collection), and DAOY cells (kind gift from Dr. Maike Glitsch, Department of Physiology, Anatomy and Genetics, University of Oxford, UK) were grown in Dulbecco's modified Eagle's medium (DMEM, Gibco/Invitrogen), while PC3 cells (kindly provided by Dr. Richard Bryant from Ludwig Institute for Cancer Research, University of Oxford, UK) were maintained in RPMI 1640 (Lonza). $Ptch1^{+/+}$, $Ptch1^{+/-}$, and $Ptch1^{-/-}$ mouse embryonic fibroblasts (MEFs, kind gift from Dr. Philip Beachy, Department of Biochemistry, Stanford University), $Sufu^{-/-}$ MEFs (kindly provided by Dr. Stephan Teglund, Department of Biosciences and Nutrition, Karolinska Institutet, Sweden), and NIH3T3 cells (ATCC) were cultured in DMEM supplemented with 1 mM sodium pyruvate (Sigma Aldrich) and 0.1 mM MEM nonessential amino acids (Sigma Aldrich).

### Hh pathway assays and production of SHH-conditioned medium

Hyperconfluent DAOY, PC3 cells, and $Sufu^{-/-}$ MEFs were starved for 24 h in serum-reduced medium (DMEM with 0.5% FBS), and subsequently, the Hh signaling pathway was stimulated using Smoothened agonist SAG (100 nM, Calbiochem) or Sonic Hedgehog (SHH)-conditioned medium (1:4). For Hh signaling inhibition, cyclopamine (Sigma Aldrich) was added to the starvation medium at a final concentration of 10 μM. DMSO (dimethyl sulfoxide) was used as a solvent and as control vehicle. $Ptch1^{+/+}$, $Ptch1^{+/-}$, and $Ptch1^{-/-}$ MEFs were assessed after 24 h of maintenance in serum-starved in serum-reduced medium. SHH-conditioned medium was generated as described previously (Cherry *et al*, 2013). After 24 h of incubation with the appropriate Hh pathway agonists, antagonist, or control vehicle, cell cultures were harvested for real-time PCR or Western blotting.

### Antibodies

The following monoclonal antibodies were used: anti-Myc (9B11, mouse, Cell Signaling Technology), anti-HA (05-904, mouse, Millipore and 16B12, Biolegend), anti-Sufu (C81H7, rabbit, Cell Signaling Technology), anti-lamin A/C (4C11, Cell Signaling Technology), and anti-vimentin (V9, mouse, Vector Laboratories). Polyclonal antibodies were used as follows: anti-Flag (F7425, rabbit, Sigma Aldrich), anti-Fbxl17 (PA5-31396, rabbit, Thermo Fisher Scientific Pierce), anti-GAPDH (MA5-15738, rabbit, Sigma Aldrich), anti-Sufu (C-15, goat, Santa Cruz Biotechnology), anti-Ki67 (SP6, rabbit, Vector Laboratories), and anti-Skp1 (kind gift from Dr. Michele Pagano at NYU Cancer Institute, New York University School of Medicine, USA). A rabbit polyclonal antibody against the following peptide of Sufu was raised Hu #347~360: CLESDS-pS-pT-AIIPHEL. To ensure the specific recognition of phosphorylated Sufu, the antibody was affinity-purified against a phosphorylated peptide and absorbed against a nonphosphorylated peptide CLESDSSTAIIPHEL.

### Liquid chromatography–tandem mass spectrometry (LC-MS/MS) analysis

Flag/Myc-tagged Fbxl17 co-immunopurified material was eluted by competition using Flag peptide. Supernatant material was subjected

to two rounds of chloroform–methanol precipitation as described (Wessel & Flugge, 1984). Samples were then resuspended in 6 M urea, 100 mM ammonium acetate pH ~8 and subjected to in-solution trypsin digestion and analysis by nano-liquid chromatography–tandem mass spectrometry as described previously (Ternette *et al*, 2013). In brief, after solubilization of the sample pellets, 3 μl of trypsin (20 ng/μl, Promega) was added to the samples diluted six times with $H_2O$ for an overnight digestion at 37°C. The samples were then desalted using a C18 SEP PAK cartridge according to the manufacturer's instructions (Waters), dried via Speed Vac centrifugation, and resuspended in 20 μl of a 2% $CH_3CN$/0.1% TFA solution. Peptides were analyzed by nano-liquid chromatography–tandem mass spectrometry (nano-LC-MS/MS) using a Nano-Acquity-UPLC (C18 column with a 75 μm × 250 mm, 1.7 μm particle size; Waters) coupled to an Orbitrap Velos tandem mass spectrometer (Thermo Scientific, Bremen, Germany) with a resolution of 70,000 full-width half maximum at mass/charge 400, top 15 precursor ion selection, and fragmentation performed in collision-induced dissociation (CID) mode. The samples were loaded in 99.5% buffer A (0.1% FA in $H_2O$). The gradient used to elute the peptides started by a 3-min isocratic gradient composed of 3% buffer B (0.1% FA in $CH_3CN$) followed by a linear gradient from 3–40% of buffer B for 60 min at a flow rate of 250 nl/min and a two washes with 97% of buffer B for 3 min. The total length of the analysis was 100 min to allow the column re-equilibration. The raw MS data were converted into Mascot generic files using MSconvert (Kessner *et al*, 2008). Searches were performed using Matrix Science software using the following parameters: The error tolerance was fixed at 20 ppm for precursor ions and at 0.5 Da for fragment ions. The enzyme used was trypsin and only one missed cleavage was allowed. MS data were searched against the human uniprot-swissprot database (UniProt_SwissProt, human 20,353 sequences) in which the false discovery rate (FDR) was estimated using a decoy database approach (Mascot) and set to 1%.

### *In vivo* and *in vitro* ubiquitylation assay

HEK293T cells were plated in 10-cm culture dishes at 80% confluence, and 24 h later, the cells were transfected with the indicated plasmids and incubated for 48 h. The cells were treated with MG132 (10 μM) for 5 h prior to collection. For Fbxl17 silencing, two rounds of siRNA transfections were performed, and the plasmids were delivered to the cells along with the siRNA duplexes in the first round of transfection. Twenty-four hours after the second round of transfection, the cells were treated with MG132 and collected 5 h later. The cells were lysed in LB and ubiquitylated Sufu was co-immunoprecipitated using either anti-HA or anti-Sufu antibody coupled to Protein G agarose beads. Polyubiquitinated forms of Sufu were detected by immunoblot using anti-Myc or anti-HA as indicated.

*In vitro* ubiquitylation assay was performed using proteins translated in a reticulocyte system, as previously done for SCF βTrcp (Dorrello *et al*, 2006). Myc-tagged Fbxl17 (318–701) and Sufu WT or Sufu K257R were *in vitro* translated using TnT Quick Coupled Transcription/Translation System reticulocyte system from Promega according to the manufacturer's instructions. After translation, Fbxl17 was mixed with Sufu WT or Sufu K257R in a volume of 40 μl containing 0.1 μM E1 (Boston Biochem), 10 ng/μl Ubch3,

10 ng/μl Ubch5c, 1 μM ubiquitin aldehyde, 2.5 μg/μl ubiquitin (Sigma), in a ubiquitylation buffer (50 mM Tris pH 7.6, 2 mM ATP, 5 mM $MgCl_2$, 0.6 mM DTT, okadaic acid 0.1 μM). The reaction mixtures were incubated at 30°C for the indicated times and analyzed by Western blot of Sufu as indicated.

### *In vitro* binding assay

HA-tagged Sufu full-length and Myc-tagged-Fbxl17 construct 318–701 was *in vitro*-transcribed and translated using TNT® Quick Coupled Transcription/Translation System (Promega) according to the manufacturer's instructions. Proteins were incubated on ice for 1 h followed by co-immunoprecipitation of Myc-tagged Fbxl17, using anti-Myc antibody, in binding buffer containing 50 mM Tris–HCl pH 7.5, 100 mM NaCl, 2 mM EDTA, 0.1% NP-40, and protease inhibitors. Reactions were stopped with Laemmli buffer and resolved by SDS–PAGE.

Sufu peptides Sufu (340–360), Sufu (340–360) phosphorylated on S352 and T353, and Sufu (351–372) (synthesized by ChinaPeptides Co., Ltd.) were conjugated to CNBr-activated Sepharose (GE Healthcare) according to the supplier's specifications. HA-tagged Sufu full length synthesized as described above and bound to Protein G–Sepharose beads was used as a positive control. Cell lysates of HEK293T cells transfected with Myc-tagged Fbxl17 were used to assess the binding to Sufu peptides using standard co-immunoprecipitation procedures.

### *In vivo* medulloblastoma model

DAOY cells transfected with either scrambled shRNA or shRNA against Fbxl17 were cultured until approximately 80–90% confluent and then resuspended in PBS for intracerebral injection.

Two cohorts ($n = 5$ per group) of male nude rats (Harlan, France), 3–4 weeks old, were injected intracerebrally in the cerebellar vermis (−11 mm posterior and 0 mm lateral to Bregma, and depth 1 mm) using a finely drawn glass microcapillary (< 100 μm tip). Animals were anaesthetized with 2–3% isoflurane in a mixture of nitrous oxide and oxygen (50%/50%) and placed in a stereotaxic frame (Stoelting Co.). A midline incision was made in the scalp and a burr hole drilled. The animals were injected with $10^4$ cells in 1 μl over a 10-min period. At the end of the injection, the needle was left in place for 2 min before being slowly removed. Following injection, the scalp wound was closed and the animals were allowed to recover from anesthesia.

### Magnetic resonance imaging

MRI was performed using a horizontal bore 9.4T magnet with a Varian DirectDrive™ (Agilent Technologies, Santa Clara, CA, USA). The animals were anesthetized with 1–2% isoflurane in 70%$N_2$/ 30%$O_2$ and a cannula positioned in the tail vein for gadolinium-DTPA (Gd-DTPA) injection. The animals were positioned in a quadrature volume transmit coil (i.d. 72 mm) with four phased-array surface receiver coils (i.d.: 40 mm, RAPID Biomedical GmbH) and placed in the magnet. Body temperature and respiration were monitored throughout and maintained at ~37°C and 50 breaths per min, respectively. Multiparametric MRI was acquired including the acquisition of $T_1$-weighted images pre- and post-Gd-DTPA injection

to assess the blood–brain barrier (BBB) integrity, and $T_2$-weighted images to determine the macroscopic changes in tissue structure, as previously described (Serres *et al*, 2014). Briefly, $T_2$-weighted images were acquired using a fast spin-echo sequence with a repetition time (TR) of 3.0 s and an effective echo time (TE) of 60 ms. Spin-echo $T_1$-weighted images (TR = 500 ms; TE = 20 ms) were acquired both before and 5 min after the injection of 100 µl Gd-DTPA (Omniscan®, GE Healthcare, UK). The matrix size and field of view were 256 × 256 and 3.5 × 3.5 cm, respectively. In all cases, a multi-slice acquisition was used, spanning the cerebellum with axial slice thickness of 1 mm.

Regions of interest (ROI) encompassing areas of visible signal change in the cerebellar vermis were drawn on both the $T_2$-weighted and post-Gd-DTPA $T_1$-weighted images, and the areas were calculated.

### Statistical analysis

Quantitative analysis of band intensity was performed using the Image J program. Data are reported as mean ± SEM. Statistical analyses were performed using GraphPad Prism (GraphPad Software, Inc.). Differences between the groups were compared using unpaired Student's *t*-test.

Affymetrix Human Gene 1.1 ST Array profiling of 285 primary medulloblastoma samples (Northcott *et al*, 2012) was obtained from Gene Expression Omnibus database (http://www.ncbi.nlm.nih.gov/geo), accession: GSE37382. Normalized, logged base 2, gene expression determined using Affymetrix Expression Console (1.1) as previously described (Northcott *et al*, 2012) was considered. One-way ANOVA and Kruskal–Wallis tests were both used to test the equality of expression values between the groups. Spearman's rank correlation test was used to test the gene expression association. Data analysis was performed using R.

**Expanded View** for this article is available online.

### Acknowledgements
This study was possible thanks to the support of Medical Research Council (MRC) Grant MC_PC_12007 to V.D'A; Cancer Research UK Grant C5255/A12678 to N.R.S.; and grant support to R.T. from the Swedish Research Council, the Swedish Cancer Society, and the Center for Innovative Medicine at Karolinska Institutet. This work was supported by a John Fell (133/075) and Wellcome Trust Grant (097813/Z/11/Z) to B.M.K. and AIRC Grant IG#14723 to L.D.M. We thank Dr. Christian Siebold for the *Sufu* cDNA and advice.

### Author contributions
MR performed most of the biochemical experiments and analyses of cell growth, and EF helped with the biochemical studies. SS, CB, and NRS designed, performed, and analyzed the animal studies. LDM and PI performed the GCPs study. BMK, RF, and M-LT performed the LC/MS analysis of Fbxl17-interacting proteins. RT and CF provided reagents, protocols, and expert advice for Hh stimulation/experiments. FMB and AB analyzed Fbxl17–Sufu–Gli1 levels in medulloblastoma. JCC provided expert advice on *in vitro* ubiquitylation experiments. VD'A and MR wrote the manuscript. VD'A coordinated the study and oversaw all experiments. All authors discussed the results, commented on and assisted in revising the manuscript.

### Conflict of interest
The authors declare that they have no conflict of interest.

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
