## [Review Process File · The EMBO Journal]

Manuscript EMBO-2015-93374

SCF (Fbx17) ubiquitylation of Sufu regulates Hedgehog signaling and medulloblastoma development

Madalina Raducu, Sebastien Serres, Alessandro Barberis, Claire Bristow, Francesca Buffa, Nicola Sibson, Marie-Laetitia Thezenas, Benedikt Kessler, Csaba Finta, Rune Toftgard and Vincenzo D'Angiolella

Corresponding author: Vincenzo D'Angiolella, CRUK/MRC Institute for Radiation Oncology

Review timeline:

Submission date:	29 October 2015
Editorial Decision:	30 November 2015
Revision received:	24 March 2016
Accepted:	29 April 2016

Editor: Hartmut Vodermaier

Transaction Report:

1st Editorial Decision

30 November 2015

Thank you for submitting your manuscript on Sufu regulation by Fbx17 to The EMBO Journal. We have now received feedback from three expert referees, whose reports are copied below for your information. I am pleased to inform you that all referees consider your findings interesting and potentially important, and therefore in principle suitable for publication in our journal. Nevertheless, they do raise a number of major issues that would need to be satisfactorily addressed before eventual acceptance. As you will see, the majority of these issues, especially in the reports of referees 1 and 3, are of technical nature, referring mainly to the conclusiveness of the biochemical data and knockdown analyses, and so would appear overall straightforward to address.

However, there are also some more significant conceptual concerns that would also need to be taken into account, especially relating to the functional significance of Sufu degradation via Fbx17. In this respect, it would be important to extend/validate at least some of the cellular assays in physiologically more relevant (cell culture) settings; as well as to better place Fbx17 into the context of Smo-mediated Hh signaling (see referee 2, as well as ref 3 point 15), ideally by including at least some follow-up investigation on how Fbx17 may be regulated by Smo activity and/or able to swiftly overcome the inhibitory effect of Sufu phosphorylation.

In this light, I would like to invite you to prepare a revised version of the manuscript, which - pending adequate answering of the discussed issues - we would be happy to consider further for publication in The EMBO Journal.

REFEREE REPORTS

Referee #1:

Through a well established and validated approach, Raducu et al. have identified the HH regulator Sufu as an interactor and a substrate of the SCF ubiquitin ligase subunit Fbx117. The authors conduct a number of studies to confirm their findings and to also show that a mutation in Sufu identified in patients affected by medulloblastoma in Gorlin syndrome, increases Sufu turnover through Fbx117-mediated ubiquitylation, leading to enhanced HH pathway activation. The authors also provide a convincing study correlating Fbx117 expression with the Shh subtype of medulloblastoma.

Overall the work is of novel and of good technical quality, it would have been significantly strengthened had the authors pursued further validation in more relevant systems. There are a number of points that need to be experimentally addressed. Much of the work is conducted in non-biologically relevant systems, although I acknowledge the inclusion of the one medulloblastoma line.

Overall immunoblotting quantification is hard to evaluate as control bands (e.g. GAPDH) are significantly overexposed. I often insist on the need to develop dilution curves to better quantify changes.

In fig. 1G first point evaluated is 6 hours, so half-life could be significantly shorter in this system. The essential role of Fbx117 for Sufu ubiquitylation could be limited to the experimental system chosen (HEK-293, fig. 2).

Sufu phosphorylation in control vs. mutant not demonstrated (fig. 3)

Standard (rescue) controls for siRNAs and shRNAs are not being provided

Fbx117 protein quantification upon silencing is not provided

Impact of Fbx117 silencing in PTCH1^{-/-} cells was quite modest (fig. 4C).

Standard representation for proliferation curves should be cell growth over time (fig 5C A, B).

Fbx117 non-targetable cDNA rescue should have been supplied.

Referee #2:

In this paper, the authors describe the identification of the SuFu protein as an interacting partner with the E3 ubiquitin ligase Fbx117 through LC-MS/MS analysis of proteins immunoprecipitated from HEK293T cells. Following up on this finding, they perform a number of studies to test the hypothesis that Fbx117 controls SuFu protein levels in response to Hh signaling. They demonstrate that Fbx117 binds directly to SuFu to promote its ubiquitylation and degradation and that this binding is inhibited by phosphorylation of SuFu and potentiated by Gli1. The data in support of these conclusions look quite good; however, quantitative analysis of the Western blots would be appropriate, along with an indication of the number of replicates performed for each assay.

The authors also investigate the functional consequences of SuFu regulation by Fbx117; they present evidence that knock down of Fbx117 abrogates Gli1 transcription factor activity in Ptch1 MEFs, in which the Hh pathway is constitutively activated, as well as in PC3 cells. In addition, they show that the response of medulloblastoma cells to Hh pathway activation can be abrogated by Fbx117.

The authors suggest that Fbx117 may play a similar role in regulating SuFu levels to that played by the Fused kinase in *Drosophila* (though they later suggest on page 15 that Fbx117 may play an analogous role in *Drosophila*). However, while it is well established that Fused activity is regulated in response to Smo activation, the authors provide no indication as to whether or how Smo activity might regulate Fbx117, save for ruling out an effect at the transcriptional level. It is quite possible that Fbx117 acts passively, binding to SuFu in response to its dephosphorylation - the authors hint at this in the Discussion when they state that "upon Hh ligand binding to Ptch1, PKA and GSK3beta are inhibited" - though they provide no justification of this statement. A better discussion of this important issue is warranted. It is also not clear to me why simply overexpressing Fbx117 is sufficient to inhibit SuFu - this implies that simply increasing the concentration of Fbx117 is sufficient to overcome the inhibitory effect of SuFu phosphorylation, but this is could be explicitly investigated.

Referee #3:

Raducu et al., propose that the SCF(Fbx17) E3 ligase complex targets Sufu for poly-ubiquitination and subsequent degradation in a Hh activation-dependent manner. They describe that phosphorylation events of Sufu regulate the interaction between Fbx17 and Sufu, and the Fbx17-mediated Sufu degradation leads to enhancement of Gli transcriptional activity. The authors further demonstrated that depletion of Fbx17 results in Sufu accumulation, leading to attenuation of medulloblastoma tumor growth. Indeed, Fbx17 and Gli mRNA expression levels are significantly elevated in clinical samples of the Shh subtype medulloblastoma. This study may provide a molecular link between Sufu alterations and cancer development/progression in various tumors, especially in medulloblastoma. The animal and clinical studies are convincing, however the biochemical data presented are somewhat premature and therefore further analyses should be carried out.

Specific comments:

1. Fig. 1B and 1C: WCL lane needs to be provided in the same panel with IP samples to show the MW of obtained Fbx17 bands in WCL are equal to the IP bands. In addition, a WB panel of IgG bands needs to be provided to indicate the same amount of IgG were used in control and Fbx17 IP reactions.
2. Fig 1E and 1G: The effects of Fbx17 knockdown are not convincing. Adding Hh stimulation may be beneficial to demonstrate a significant Sufu stabilization following Fbx17 depletion.
3. Fig 1E, 1G, 2A, 4A, 4E, 5A-5C and 6E: Fbx17 blots should be provided to show relevant Fbx17 knockdown at protein level. This should be relatively straight forward as the Fbx17 antibody for WB analysis has been validated in Fig. 1B and 1C.
4. Fig. 1G: It would be helpful to present the data in a graph by quantifying the band intensities of three independent experiments.
5. Fig. 2A: WB panels of input (HA, Fbx17 and GAPDH blots) should be included.
6. Fig. 2: In vitro ubiquitination assay needs to be performed to prove that SCF(Fbx17) can directly transfer polyubiquitin chain to Sufu on K257.
7. Fig. 3A: The phosphorylation mimetic S to D mutant should be added in this analysis as performed in Fig. 2B.
8. Fig. 3B: It would be better to include Flag-Gli1 in the assay as performed in Fig. 2A.
9. Fig 3D: It is difficult to interpret the data as the panels are confusing and not clearly or appropriately labeled.
10. Fig 3E: The strong polyubiquitination band, which is observed in the last lane, is not supposed to appear, as Fbx17 is absent in this lane. Is it a contaminating band derived from polyubiquitinated Gli1 protein? The authors need to exclude this possibility.
11. Fig. 5B: Sufu and Fbx17 blots need to be provided.
12. Fig. 6: Did the authors examine the frequency of Sufu mutations at the S352, S342 and S346 phosphorylation sites in the cohorts of the 285 medulloblastoma clinical samples?
13. EV3A-3B: Sufu Δ 350-425 mutants should be included for the mapping. As authors mention that Fbx1 proteins recognize a larger surface rather than a linear degron motif on substrates, it is critical to use a minimal deletion mutant. Extensive deletion might cause mis-folding, leading to a nonspecific interaction.
14. EV3C: S342/346 mutant need to be included as a positive control to set the criteria of what extent of enhancement is significant.

15. The authors propose the failsafe model through which PKA and GSK3beta phosphorylate Sufu to prevent unscheduled Sufu degradation by Fbx117. However, upon Hh activation, the cells require a swift response to transmit the Hh signals into nucleus. Regarding the dephosphorylation of Sufu, which molecular mechanisms are involved in efficiently releasing the failsafe system?

1st Revision - authors' response

24 March 2016

Please find below our point-by-point answers to referees:

Referee #1:

Through a well established and validated approach, Raducu et al. have identified the HH regulator Sufu as an interactor and a substrate of the SCF ubiquitin ligase subunit Fbx117. The authors conduct a number of studies to confirm their findings and to also show that a mutation in Sufu identified in patients affected by medulloblastoma in Gorlin syndrome, increases Sufu turnover through Fbx117-mediated ubiquitylation, leading to enhanced HH pathway activation. The authors also provide a convincing study correlating Fbx117 expression with the Shh subtype of medulloblastoma.

Overall the work is of novel and of good technical quality, it would have been significantly strengthened had the authors pursued further validation in more relevant systems. There are a number of points that need to be experimentally addressed.

1. Much of the work is conducted in non-biologically relevant systems, although I acknowledge the inclusion of the one medulloblastoma line.

Answer 1:

One of the main limitations in the Hedgehog signaling field is the poor availability of cell lines, which maintain Hedgehog pathway active or in which the pathway can be modulated by treatments with agonists/antagonists. There are controversial opinions about the cell systems suitable for studying molecular regulation of Hedgehog signaling. For this reason, we conducted an extensive validation of pathway activation in DAOY in Fig. EV7. In addition to this, we have confirmed the functional role of Fbx117-Sufu axis in PC3 cell lines (Fig EV5 and 6), which do not rely on a ligand dependent mechanism of Hedgehog pathway activation but show regulation of Sufu by proteolysis and transcriptional regulation of Gli, in accordance to previous findings (Zhang et al, 2007). Mouse embryonic fibroblasts (MEFs) deficient in either Ptch1 (a major negative regulator of Hedgehog signaling) or Sufu have been extensively used in Hedgehog signaling to assess pathway activation. These have been used in our study in Figure 4 and 7F.

Most importantly, to establish a role for Fbx117 in SHH medulloblastoma development, we show in Fig EV 10 A, B, C and D that Fbx117 has an important role in the proliferation of Granule Cell Progenitors, the cells of origin of SHH medulloblastoma (Marino et al, 2000; Oliver et al, 2005; Schuller et al, 2008; Wechsler-Reya & Scott, 1999).

2. Overall immunoblotting quantification is hard to evaluate as control bands (e.g. GAPDH) are significantly overexposed. I often insist on the need to develop dilution curves to better quantify changes.

Answer 2:

We do agree that some of the immunoblots contained oversaturated loading control bands and we apologize for that. To overcome this issue, some of these immunoblots were repeated and new panels containing less exposed loading control bands were included in Fig 1B, 1C, 1E, 2B and 5A. Relative quantifications have been modified accordingly without substantial changes in the significance of findings.

3. In fig. 1G first point evaluated is 6 hours, so half-life could be significantly shorter in this system

Answer 3:

Fbx117 levels could significantly change in different cell lines (<http://www.broadinstitute.org/ccle/home>) and affect Sufu half-life accordingly. Thus, it is difficult to make a general statement regarding Sufu half-life. Of note, in the experiments performed in Fig 1G and Fig7E siRNA of Fbx117 with two different oligos induced a significant increase of Sufu half-life.

4. The essential role of Fbx117 for Sufu ubiquitylation could be limited to the experimental system chosen (HEK-293, fig. 2)

Answer 4:

We have performed ubiquitylation assay of Sufu in a more relevant cell system (DAOY cells) and this is shown now in Figure 2A. Absent polyubiquitylated species of Sufu upon Fbx117 depletion reinforce the essential role for Fbx117 in Sufu ubiquitylation in DAOY medulloblastoma cancer cell lines.

5. Sufu phosphorylation in control vs. mutant not demonstrated (fig. 3)

Answer 5:

Sufu was found to be phosphorylated on S352 in the following previous publication using LC/MS (Hsu et al, 2011). We have performed LC/MS analysis of Sufu secondary modifications in FigEV4A, where it is shown that the corresponding peptide could contain numerous modifications. Furthermore, we raised an antibody against a peptide 347-360 of Sufu containing S352 and T353 phosphorylated. We detected Sufu phosphorylation on Sufu WT but not on a mutant of Sufu where S352 and T353 were substituted to alanine (Fig.3C). This shows that Sufu is phosphorylated *in vivo* on S352/T353.

6. Standard (rescue) controls for siRNAs and shRNAs are not being provided
Fbx117 protein quantification upon silencing is not provided

Answer 6:

Rescue of cell proliferation and Sufu protein levels is observed after siRNA of Fbx117 and rescue upon expressing Fbx117 full length in DAOY cells (Fig 5A, 5C, Fig EV8A and B) and PC3 cells (Fig EV5C-E). Protein levels of Fbx117 are presented in Fig 5C and Fig EV5D. Due to antibody limitation we measured the extent of Fbx117 siRNA by QPCR in Fig 5B and E and Fig EV 5F.

7. Impact of Fbx117 silencing in PTCH1^{-/-} cells was quite modest (fig. 4C).

Answer 7:

Fbx117 siRNA induces a reduction of Gli1 mRNA, which is highly significant ($p < 0.0005$).

8. Standard representation for proliferation curves should be cell growth over time (fig 5C A, B).

Answer 8:

Given the nature of growth of DAOY cells we perform few determinations of cell numbers, which are not well represented using cell growth over time. This representation has been used in previous publications (McKee et al, 2012). The effect of Fbx117 on cell proliferation has been followed with different methods, which confirm the validity of findings:

1. Measurements of relative cell proliferation
2. The use of an orthotopic rat model of medulloblastoma to monitor cell growth at different time points by MRI scan, which gives accurate determination of tumour volume and size superior to bioluminescence.
3. Using marker of cell proliferation such as Ki67.
4. Using BrdU incorporation in GCPs.

9. Fbx117 non-targetable cDNA rescue should have been supplied.

Answer 9:

Rescue using cDNA for Fbx117 is provided in: Fig 5A and C, Fig EV8A and B and in Fig EV5C-E.

Referee #2:

1. In this paper, the authors describe the identification of the SuFu protein as an interacting partner with the E3 ubiquitin ligase Fbx117 through LC-MS/MS analysis of proteins immunoprecipitated from HEK293T cells. Following up on this finding, they perform a number of studies to test the hypothesis that Fbx117 controls SuFu protein levels in response to Hh signaling. They demonstrate that Fbx117 binds directly to SuFu to promote its ubiquitylation and degradation and that this binding is inhibited by phosphorylation of SuFu and potentiated by Gli1. The data in support of these conclusions look quite good; however, quantitative analysis of the Western blots would be appropriate, along with an indication of the number of replicates performed for each assay.

Answer 1:

Quantification of immunoblots was performed for Fig 1E (see Fig EV1A), Fig 1F (see Fig EV1B), Fig 1G (see Fig EV1E), Fig 5C (see Fig EV8A), Fig 5F (see Fig EV8B), Fig 6B (see Fig EV9A), Fig EV 5D (see Fig EV5E) and Fig EV6B (see Fig EV6C). The number of replicates performed for these experiments was added in the corresponding figure legend.

2. The authors also investigate the functional consequences of SuFu regulation by Fbx117; they present evidence that knock down of Fbx117 abrogates Gli1 transcription factor activity in Ptch1 MEFs, in which the Hh pathway is constitutively activated, as well as in PC3 cells. In addition, they show that the response of medulloblastoma cells to Hh pathway activation can be abrogated by Fbx117.

The authors suggest that Fbx117 may play a similar role in regulating SuFu levels to that played by the Fused kinase in *Drosophila* (though they later suggest on page 15 that Fbx117 may play an analogous role in *Drosophila*). However, while it is well established that Fused activity is regulated in response to Smo activation, the authors provide no indication as to whether or how Smo activity might regulate Fbx117, save for ruling out an effect at the transcriptional level. It is quite possible that Fbx117 acts passively, binding to SuFu in response to its dephosphorylation - the authors hint at this in the Discussion when they state that "upon Hh ligand binding to Ptch1, PKA and GSK3beta are inhibited" - though they provide no justification of this statement. A better discussion of this important issue is warranted.

Answer 2:

We have now extended the discussion to clarify this point. We do agree that Fbx117 acts passively after Sufu dephosphorylation. It has been reported that Sufu phosphorylation promotes its retention within the cilium (Chen et al, 2011) thus establishing a spatio-temporal determinant of Sufu dephosphorylation and ubiquitylation after pathway activation.

However, the regulation of Fbx117-Sufu axis is likely to be complex and different in the diverse tissue and models analyzed. For instance it is tantalizing to speculate that cells with a functional cilium could also regulate Fbx117. These studies could not be undertaken in the current work, which focuses on Sufu regulation by Fbx117.

We don't think that Fbx117 is playing an analogue role in *Drosophila* in Sufu degradation since the role of Sufu is substantially different in this model system and Fbx117 has a little to poor sequence conservation to CG31633 (the postulated homologue). However, the essential role of CG31633 in *Drosophila* development emphasizes the importance of Fbx117 during embryogenesis. We hope that our comments will stimulate studies on CG31633, which will clarify its role in this fascinating model system. We have changed the discussion to clarify our statement.

3. It is also not clear to me why simply overexpressing Fbx117 is sufficient to inhibit SuFu - this implies that simply increasing the concentration of Fbx117 is sufficient to overcome the inhibitory effect of SuFu phosphorylation, but this is could be explicitly investigated.

Answer 3:

From previous literature it is clear that Sufu dephosphorylation facilitate its relocalization outside the cilium (Chen et al, 2011). Our data are in accordance to a passive model whereby the dephosphorylated fraction of Sufu is polyubiquitylated by Fbx117. This mechanism is similar but opposite to the regulation of Gli2 and Gli3 operated by SCF^{bTrep} (Bhatia et al, 2006; Wang & Li, 2006). In both cases proper activation of Hh signaling needs reversal of phosphorylation mediated by PKA and GSK3b. Of note, while dephosphorylation is a central mechanism to proper Hh signaling (Eisner et al, 2015), the players mediating Sufu dephosphorylation are unknown.

Sufu can inhibit Hedgehog signaling also in the absence of cilia (Jia et al, 2009). In cancer cell lines, lacking cilia, a pool of Sufu, which is not phosphorylated could be present, due to constitutive pathway activation. In PC3 and DAOY activating mutation in Hedgehog signaling components could generate a pool of Sufu readily degradable by Fbx117. This could explain the effect of Fbx117 overexpression on Sufu levels. Importantly, in MEFs with two copies of *Ptch1*, in which the pathway is inactive, siRNA of Fbx117 does not induce alteration in Sufu levels.

Referee #3:

Raducu et al., propose that the SCF(Fbx117) E3 ligase complex targets Sufu for poly-ubiquitination and subsequent degradation in a Hh activation-dependent manner. They describe that phosphorylation events of Sufu regulate the interaction between Fbx117 and Sufu, and the Fbx117-mediated Sufu degradation leads to enhancement of Gli transcriptional activity. The authors further demonstrated that depletion of Fbx117 results in Sufu accumulation, leading to attenuation of medulloblastoma tumor growth. Indeed, Fbx117 and Gli mRNA expression levels are significantly elevated in clinical samples of the Shh subtype medulloblastoma. This study may provide a molecular link between Sufu alterations and cancer development/progression in various tumors, especially in medulloblastoma. The animal and clinical studies are convincing, however the biochemical data presented are somewhat premature and therefore further analyses should be carried out.

Specific comments:

1. Fig. 1B and 1C: WCL lane needs to be provided in the same panel with IP samples to show the MW of obtained Fbx117 bands in WCL are equal to the IP bands. In addition, a WB panel of IgG bands needs to be provided to indicate the same amount of IgG were used in control and Fbx117 IP reactions.

Answer 1:

This has been provided as requested in the new Fig 2B and C. It is important to note that the commercially available antibody that we use recognize Fbx117 only after enriching its levels by immunoprecipitation of Sufu. As a reference we have immunoprecipitated exogenous Fbx117 which migrates at the same molecular weight as the endogenous. A panel containing IgG bands was also introduced to indicate the amount of IgG used for each immunoprecipitation.

2. Fig 1E and 1G: The effects of Fbx117 knockdown are not convincing. Adding Hh stimulation may be beneficial to demonstrate a significant Sufu stabilization following Fbx117 depletion.

Answer 2:

After Hedgehog stimulation using SAG (Fig 1E, and quantified in Fig EV1A) the effects were more significant and panels have been changed accordingly. We do agree that a proper activation of Hedgehog signaling is a limiting factor for Sufu stabilization upon Fbx117 depletion. A better effect can be observed in Fig 4A, where *Ptch1*-depleted cells, with a constitutive Hh signaling activation, were used.

3. Fig 1E, 1G, 2A, 4A, 4E, 5A-5C and 6E: Fbx117 blots should be provided to show relevant Fbx117 knockdown at protein level. This should be relatively straight forward as the Fbx117 antibody for WB analysis has been validated in Fig. 1B and 1C.

Answer 3:

Please note (Fig 1B and C) that the antibody against Fbx117 did not work on endogenous protein but only after enriching by immunoprecipitation of Sufu. We have performed qPCR of Fbx117 for the experiments in Fig 1E (see Fig 1D), 1G (see Fig EV1F), 2A and B (see Fig 2C), 4A (see Fig 4B), 4E (see EV 4B), 5A (see Fig 5B), Fig 5D (see Fig 5E), Fig 6B (see Fig 6A), Fig EV5A (see Fig EV5F) and Fig EV6A (see Fig EV 6D). All the qPCR analysis show a drastic reduction of Fbx117 mRNA in all cases.

4. Fig. 1G: It would be helpful to present the data in a graph by quantifying the band intensities of three independent experiments.

Answer 4:

This has been presented in figure EV1E.

5. Fig. 2A: WB panels of input (HA, Fbx117 and GAPDH blots) should be included.

Answer 5:

In the new figure 2B, Western blot of Sufu and GAPDH have been included, and also QPCR of Fbx117 has been reported in Fig 2C.

6. Fig. 2: In vitro ubiquitination assay needs to be performed to prove that SCF(Fbx117) can directly transfer polyubiquitin chain to Sufu on K257.

Answer 6:

In vitro ubiquitylation assay of Sufu WT and Sufu K257R mutant was introduced in Fig EV2C.

7. Fig. 3A: The phosphorylation mimetic S to D mutant should be added in this analysis as performed in Fig. 2B.

Answer 7:

This has been added in Figure EV3D

8. Fig. 3B: It would be better to include Flag-Gli1 in the assay as performed in Fig. 2A.

Answer 8:

A new western blot has been introduced in Fig EV3E.

9. Fig 3D: It is difficult to interpret the data as the panels are confusing and not clearly or appropriately labeled.

Answer 9:

We apologize for this and have modified the figure for improved clarity.

10. Fig 3E: The strong polyubiquitination band, which is observed in the last lane, is not supposed to appear, as Fbx117 is absent in this lane. Is it a contaminating band derived from polyubiquitinated Gli1 protein? The authors need to exclude this possibility.

Answer 10:

This band derives from polyubiquitylation of Sufu mediated by endogenous Fbx117. Sufu polyubiquitylation by endogenous Fbx117 can also be observed in Fig 2A and B.

11. Fig. 5B: Sufu and Fbx117 blots need to be provided.

Answer 11:

As we mentioned earlier, due to antibody limitations, levels of endogenous Fbx117 could not be assessed by Western blot. For Fig 5B (now Fig 5D), qPCR showing downregulation in Fbx117 mRNA levels has been added in Fig 5E. Also, a representative image of three independent experiments along with the corresponding relative quantification has been added for Sufu protein in Fig EV8A and Fig EV8B.

12. Fig. 6: Did the authors examine the frequency of Sufu mutations at the S352, S342 and S346 phosphorylation sites in the cohorts of the 285 medulloblastoma clinical samples?

Answer 12:

This cohort does not contain information on Sufu mutations.

13. EV3A-3B: Sufu Δ 350-425 mutants should be included for the mapping. As authors mention that Fbx1 proteins recognize a larger surface rather than a linear degron motif on substrates, it is critical to use a minimal deletion mutant. Extensive deletion might cause mis-folding, leading to a nonspecific interaction.

Answer 13:

We have included the mutant in Figure EV3C, which binds Fbx117.

14. EV3C: S342/346 mutant need to be included as a positive control to set the criteria of what extent of enhancement is significant.

Answer 14:

Provided in figure EV3E.

15. The authors propose the failsafe model through which PKA and GSK3beta phosphorylate Sufu to prevent unscheduled Sufu degradation by Fbx117. However, upon Hh activation, the cells require a swift response to transmit the Hh signals into nucleus. Regarding the dephosphorylation of Sufu, which molecular mechanisms are involved in efficiently releasing the failsafe system?

Answer 15:

From previous literature it is clear that Sufu dephosphorylation facilitate its relocalization outside the cilium(Chen et al, 2011). Our data are in accordance to a passive model whereby the dephosphorylated fraction of Sufu is polyubiquitylated by Fbx117. This mechanism is similar but opposite to the regulation of Gli2 and Gli3 operated by SCF^{bTrcp}. In both cases proper activation of Hh signaling needs reversal of phosphorylation mediated by PKA and GSK3b. Of note, while dephosphorylation is central to proper Hh signaling (Eisner et al, 2015), the mechanisms underlying Sufu dephosphorylation are unknown.

References:

- Bhatia N, Thiyagarajan S, Elcheva I, Saleem M, Dlugosz A, Mukhtar H, Spiegelman VS (2006) Gli2 is targeted for ubiquitination and degradation by beta-TrCP ubiquitin ligase. *J Biol Chem* **281**: 19320-19326
- Chen Y, Yue S, Xie L, Pu XH, Jin T, Cheng SY (2011) Dual Phosphorylation of suppressor of fused (Sufu) by PKA and GSK3beta regulates its stability and localization in the primary cilium. *J Biol Chem* **286**: 13502-13511
- Eisner A, Pazyra-Murphy MF, Durresti E, Zhou P, Zhao X, Chadwick EC, Xu PX, Hillman RT, Scott MP, Greenberg ME, Segal RA (2015) The Eya1 phosphatase promotes Shh signaling during hindbrain development and oncogenesis. *Developmental cell* **33**: 22-35
- Hsu PP, Kang SA, Rameseder J, Zhang Y, Ottina KA, Lim D, Peterson TR, Choi Y, Gray NS, Yaffe MB, Marto JA, Sabatini DM (2011) The mTOR-regulated phosphoproteome reveals a mechanism of mTORC1-mediated inhibition of growth factor signaling. *Science* **332**: 1317-1322
- Jia J, Kolterud A, Zeng H, Hoover A, Teglund S, Toftgard R, Liu A (2009) Suppressor of Fused inhibits mammalian Hedgehog signaling in the absence of cilia. *Developmental biology* **330**: 452-460
- Marino S, Vooijs M, van Der Gulden H, Jonkers J, Berns A (2000) Induction of medulloblastomas in p53-null mutant mice by somatic inactivation of Rb in the external granular layer cells of the cerebellum. *Genes Dev* **14**: 994-1004
- McKee CM, Xu D, Cao Y, Kabraji S, Allen D, Kersemans V, Beech J, Smart S, Hamdy F, Ishkanian A, Sykes J, Pintile M, Milosevic M, van der Kwast T, Zafarana G, Ramnarine VR, Jurisica I, Malloff C, Lam W, Bristow RG, Muschel RJ (2012) Protease nexin 1 inhibits hedgehog signaling in prostate adenocarcinoma. *The Journal of clinical investigation* **122**: 4025-4036
- Oliver TG, Read TA, Kessler JD, Mehmeti A, Wells JF, Huynh TT, Lin SM, Wechsler-Reya RJ (2005) Loss of patched and disruption of granule cell development in a pre-neoplastic stage of medulloblastoma. *Development* **132**: 2425-2439
- Schuller U, Heine VM, Mao J, Kho AT, Dillon AK, Han YG, Huillard E, Sun T, Ligon AH, Qian Y, Ma Q, Alvarez-Buylla A, McMahon AP, Rowitch DH, Ligon KL (2008) Acquisition of granule

neuron precursor identity is a critical determinant of progenitor cell competence to form Shh-induced medulloblastoma. *Cancer cell* **14**: 123-134

Wang B, Li Y (2006) Evidence for the direct involvement of β TrCP in Gli3 protein processing. *Proceedings of the National Academy of Sciences of the United States of America* **103**: 33-38

Wechsler-Reya RJ, Scott MP (1999) Control of neuronal precursor proliferation in the cerebellum by Sonic Hedgehog. *Neuron* **22**: 103-114

Zhang J, Lipinski R, Shaw A, Gipp J, Bushman W (2007) Lack of demonstrable autocrine hedgehog signaling in human prostate cancer cell lines. *The Journal of urology* **177**: 1179-1185

2nd Editorial Decision

29 April 2016

Thank you for submitting your revised manuscript for our consideration by the EMBO Journal. It has now been seen again by two of the original referee (see comments below). I am please to inform you that both of them are satisfied with the responses and revisions and have no further objections towards publication. We are therefore happy to accept your manuscript in principle at this stage.

REFeree REPORTS

Referee #2:

The authors have addressed the points raised in my original review adequately.

Referee #3:

The authors have improved the manuscript. This is suitable for publication.

Corresponding Author Name: Vincenzo D'Angiolella
 Manuscript Number: EMBOJ-2015-93374